# Carbon hollow fiber membranes for a molecular sieve with precise-cutoff ultramicropores for superior hydrogen separation

Linfeng Lei[1], Fengjiao Pan[2], Arne Lindbråthen[1], Xiangping Zhang [2], Magne Hillestad[1], Yi Nie[2], Lu Bai[2], Xuezhong He [1,3✉] & Michael D. Guiver [4✉]

Carbon molecular sieve (CMS) membranes with rigid and uniform pore structures are ideal candidates for high temperature- and pressure-demanded separations, such as hydrogen purification from the steam methane reforming process. Here, we report a facile and scalable method for the fabrication of cellulose-based asymmetric carbon hollow fiber membranes (CHFMs) with ultramicropores of 3–4 Å for superior $H_2$ separation. The membrane fabrication process does not require complex pretreatments to avoid pore collapse before the carbonization of cellulose precursors. A $H_2/CO_2$ selectivity of 83.9 at 130 °C ($H_2/N_2$ selectivity of >800, $H_2/CH_4$ selectivity of >5700) demonstrates that the membrane provides a precise cutoff to discriminate between small gas molecules ($H_2$) and larger gas molecules. In addition, the membrane exhibits superior mixed gas separation performances combined with water vapor- and high pressure-resistant stability. The present approach for the fabrication of high-performance CMS membranes derived from cellulose precursors opens a new avenue for $H_2$-related separations.

[1] Department of Chemical Engineering, Norwegian University of Science and Technology, 7491 Trondheim, Norway. [2] Beijing Key Laboratory of Ionic Liquids Clean Process, Institute of Process Engineering, Chinese Academy of Sciences, P.O. Box 353, 100190 Beijing, China. [3] Department of Chemical Engineering, Guangdong Technion Israel Institute of Technology (GTIIT), 241 Daxue Road, 515063 Shantou, China. [4] State Key Laboratory of Engines, School of Mechanical Engineering, and Collaborative Innovation Center of Chemical Science and Engineering (Tianjin), Tianjin University, 300072 Tianjin, China. ✉email: xuezhong.he@gtiit.edu.cn; michael.guiver@outlook.com

Hydrogen production from natural gas is considered as one of the most promising and large-scale technologies for the implementation of the hydrogen economy, with respect to a low-carbon energy future and the reduction of greenhouse gas emissions. In comparison with conventional hydrogen purification technologies such as pressure swing adsorption (PSA) and fractional/cryogenic distillation, membrane-based separation technology is currently considered as a promising alternative owing to its lower investment cost, intrinsic higher energy efficiency, and environmental friendliness[1,2]. Various membrane materials such as polymeric membranes[3,4], inorganic-based membranes like graphene oxide (GO)[5], $MoS_2$[6], zeolite imidazolate framework (ZIF)[7,8], and metal-organic frameworks (MOFs)[9,10] have been developed for $H_2/CO_2$ separation. However, achieving some commercially viable membranes for $H_2$ purification is still challenging, either due to a low separation performance, or complex preparation processes (high cost), or limited stability under adverse conditions (e.g., high temperature and pressure in the steam methane reforming process). Carbon molecular sieve (CMS) membranes have rigid pore structures and are fabricated by controlled carbonization of polymeric precursors at high temperature. CMS membranes are promising candidates as temperature- and pressure-resistant materials when fabricated into hollow fibers suitable for membranes modules[11,12]. The bimodal pore structure of CMS membranes, which comprises ultramicropores and micropores provides favorable gas selectivity in $H_2$-related separations such as $H_2/CH_4$[12] and $H_2/C_2H_4$[13]. However, due to the strong adsorption between the carbon surface and $CO_2$ molecules, relatively low $H_2/CO_2$ selectivities of <10 were usually reported[12,14,15]. Tailoring the ultramicropores in CMS membranes would provide a powerful approach to obtain CMS membranes with higher selectivities for $H_2/CO_2$ separation. Recently, Ma et al. reported an $H_2$-assisted method to create "mid-sized" ultramicropores (5–7 Å) in CMS membranes by introducing $H_2$ into the carbonization environment[11]. The introduction of $H_2$ during the carbonization process was found to inhibit aromatization during thermal decomposition of the polymer network, resulting in a structure with wider ultramicropores compared with the CMS membranes made using argon atmosphere. Introducing an additional thermal treatment step for the freshly-prepared CMS membranes at a temperature range of 90–250 °C, referred to as "hyperaging treatment" to accelerate aging, was shown to create smaller ultramicropores as reported by Qiu et al.[13]. However, the CMS membranes reported so far still present relatively larger ultramicropores, which does not allow precise gas sieving between $H_2$ and $CO_2$. In addition to exploring carbonization conditions suitable for efficient $H_2/CO_2$ separations with CMS membranes, the selection of suitable polymeric precursors is a crucial factor in achieving the desired membrane separation performance since precursor structures significantly affect the pore structures and properties of the derived CMS membranes[16]. It has been reported that increasing the doping content of heteroatoms, such as N, into carbon materials could enhance the $CO_2$ sorption capabilities, thereby providing highly $CO_2$-selective materials[17,18]. To et al.[17] demonstrated Henry's Law selectivity of $CO_2/N_2$ for N-doped porous carbons increased from 9 to 124 when the N content was increased from 3.2 to 5.8 wt.%. Similarly, Yang et al.[18] reported that the diffusion of $CO_2$ molecules through the membrane was improved by N- and F-containing nanodomains. As a corollary, it is inferred that CMS materials with a low heteroatom-content should reduce $CO_2$ sorption, and may provide an approach to achieve highly $H_2/CO_2$ selective carbon membranes. Cellulose, an abundant renewable and low-cost biomaterial, containing only C, H, and O atoms, has recently exhibited particularly promising potential as an effective precursor for CMS membranes with high gas selectivities[19,20]. Moreover, microcrystalline cellulose (MCC), which has high crystallinity, is advantageous for forming CMS membranes

with a more ordered graphitic structure[21], and may thus provide high selectivity by molecular sieving. In addition, it was found that the cellulose-derived CMS membrane has shown stable performance for $O_2/N_2$ separation under ~80% relative humidity because of its high hydrophilicity that allows water vapor to permeate the membrane easily[20]. The negligible effect on pore blockage caused by water vapor suggests that cellulose-based CMS membranes have great potential for $H_2$ purification in the steam methane reforming process, where water vapor is normally present.

On the other hand, the development of self-supported asymmetric CMS membranes (i.e., a very thin selective layer integrally supported on a porous substrate) is highly desired to achieve higher gas permeances as the required membrane area is inversely proportional to gas permeance. However, maintaining the asymmetric structural morphology during carbonization is still an ongoing challenge. During the carbon matrix thermal formation process, the porous layer of asymmetric precursors is prone to collapsing and forming a relatively denser layer, which leads to a deterioration of gas permeance[22,23]. An additional pre-treatment process often referred to as the crosslinking step[24,25] that normally requires a chemical treatment or the coating of a new physical layer, is needed to "lock in" the asymmetric structure. However, in addition to increasing the fabrication complexity and potentially introducing defects into the selective layer, the supplementary associated step may also account for ~40% cost increment of the overall membrane production process[22]. Thus, the preparation of asymmetric CMS membranes without a complex crosslinking step is pragmatically significant.

Although cellulose membranes and its derived CMS membranes have been reported in the literature[19,20,26], asymmetric cellulose-based carbon hollow fiber membranes have never been reported for gas separation. Cellulose has strong inter- and intrachain hydrogen bonds, which prevents solubility in most common solvents. Only a few solvents, such as N-methylmorpholine-N-oxide (NMMO), ionic liquids, and inorganic salts, are effective in breaking the extensive hydrogen-bonding network[27,28]. Nevertheless, obtaining accurate cellulose/solvent/non-solvent ternary phase diagrams are still challenging because of the enormous viscosity of the systems. Thus, it is difficult to control the membrane formation phase inversion process for such a system, which ultimately determines the structure and morphology of regenerated hollow fiber cellulose precursors. Furthermore, it has been found that water-filled cellulose membranes experience morphology collapse when dried directly from water[26]. Accordingly, a proper drying process to maintain the formed pore structure is required.

Here, we demonstrate the preparation of asymmetric cellulose hollow fiber precursor membranes by tuning the coagulation temperature from the MCC/ionic liquid/water system. The water-wetted cellulose membranes were dried using an exchange process with non-solvent (water–isopropanol–n-hexane) to maintain the obtained asymmetric structure of cellulose precursors. This protocol avoids a crosslinking pre-treatment to prevent pore collapse during the carbonization process. Afterward, the CHFMs with the asymmetric structure were fabricated by tuning the final carbonization temperature to exhibit superior separation performance in terms of $H_2$ permeance and $H_2/CO_2$, $H_2/N_2$, and $H_2/CH_4$ selectivities and also provide a high pressure-resistant capability, which is usually required for industrial $H_2$ purification processes in steam methane reforming plants.

## Results

**Fabrication of asymmetric carbon hollow fiber membranes.** The asymmetric cellulose hollow fiber precursors were fabricated by a dry-wet spinning process as illustrated in Fig. 1a. It must be

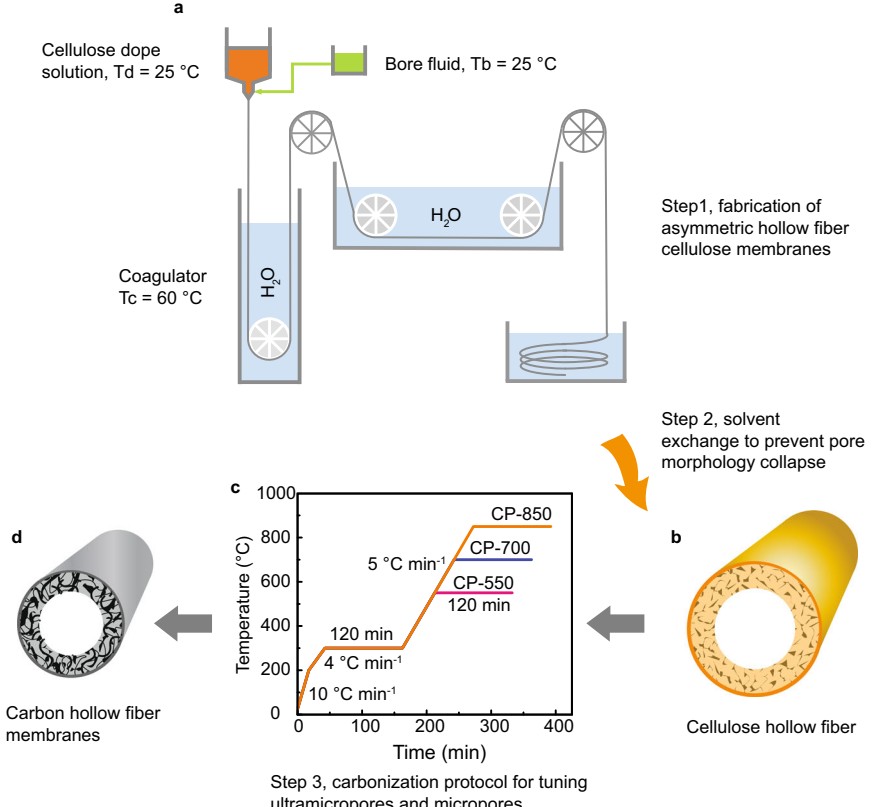

**Fig. 1 Preparation of carbon hollow fiber membranes (CHFMs). a** Schematic of the fabrication process for asymmetric cellulose hollow fibers by the dry-wet spinning process with controlled spinning temperatures. Temperature of dope solution (Td), bore solution (Tb), and coagulator (Tc) are 25, 25, and 60 °C, respectively. **b** Schematic of dried cellulose hollow fiber precursors. **c** Carbonization protocols (CPs) for the fabrication of CHFMs. **d** Schematic of asymmetric CHFMs. The key step 1 is the fabrication of asymmetric cellulose hollow fibers by controlling the coagulation temperature at >45 °C (60 °C was used in this work); step 2 is non-solvent exchange using lower surface tension solvents (isopropanol, *n*-hexane) to remove residual water inside hollow fibers before drying to prevent pore morphology collapse; step 3 is tuning the ultramicropore and micropore structure of carbon membranes by changing the final carbonization temperature from 550 to 850 °C.

noted that the asymmetric structure of cellulose membranes with an integral top dense layer supported by a porous layer can be formed only when the coagulation bath temperature is above 45 °C (for details see Supplementary Note 1 and Supplementary Fig. 1). The present work is the first to report the preparation of asymmetric cellulose hollow fibers directly from a cellulose/ionic liquid casting dope system. Moreover, to prevent pore collapse during the fiber drying process, a simple non-solvent exchange step was employed (water–isopropanol–*n*-hexane). Compared with hollow fibers dried directly from a water bath with dense structure (Supplementary Fig. 4 and Supplementary Note 2), the cellulose hollow fibers processed using the anti-pore collapse treatment exhibit asymmetric structural morphology, including a porous inner layer, a middle layer rich in macrovoids and a dense outer layer (Supplementary Fig. 5a). By following different carbonization protocols as depicted in Fig. 1c, the asymmetric structure was well maintained with an outer selective layer of ca. 3 μm and an integral porous inner support layer (see the SEM images in Fig. 2a, b). The prepared carbon membranes show similar skin layer thickness, determined mainly by precursor dimensions, which can be adjusted by tuning spinning parameters such as coagulation temperature and take-up speed. To the best of our knowledge, this is the first report of the fabrication of asymmetric CHFMs directly from cellulose hollow fiber precursors. Moreover, it should be noted that the prepared carbon membranes present good mechanical flexibility with a bend radius of <1.5 cm, as indicated in the inset of Fig. 2a. The Young's modulus of prepared CHFMs measured by nanoindentation tests

is summarized in Supplementary Table 2 and load–displacement curves are shown in Supplementary Fig. 7. Like other carbonized carbon materials[29,30], the CHFM at the lowest carbonization temperature, CHFM-550, exhibits the deepest displacement by the indentation, leading to the lowest hardness (0.31 GPa) and Young's modulus (2.07 GPa). As the carbonization temperature increases, the hardness and Young's modulus increase to 1.30 and 7.53 GPa, respectively. The enhanced hardness and modulus can be attributed to the change of internal structure by raising carbonization temperature, such as the increase of $sp^2$-hybridized bonds in carbon[29].

**Structure of carbon hollow fiber membranes**. To achieve both superior $H_2/CO_2$ selectivity and high $H_2$ permeance, three carbonization temperatures (550, 700, and 850 °C) were investigated, denoted as CHFM-550, CHFM-700, and CHFM-850, respectively. The XRD patterns (Fig. 2c) for these CHFMs reveal the characteristic peak for $2\theta$ at around 24°, which corresponds to the (002) plane of the graphite phase ($sp^2$ carbon). Moreover, a peak shift to a higher $2\theta$ indicates that the average inter-plane distance ($d_{002}$) decreases from 3.78 to 3.50 Å when the carbonization temperature increases from 550 to 850 °C. This indicates that the carbon membranes prepared at higher carbonization temperatures tend to form graphitic carbon (~3.4 Å) with a more ordered graphitic structure and smaller pores. The pore size distribution (PSD, Fig. 2d) in the range of 3–10 Å obtained by $CO_2$ physisorption at 0 °C confirms the narrowing of the pore width of

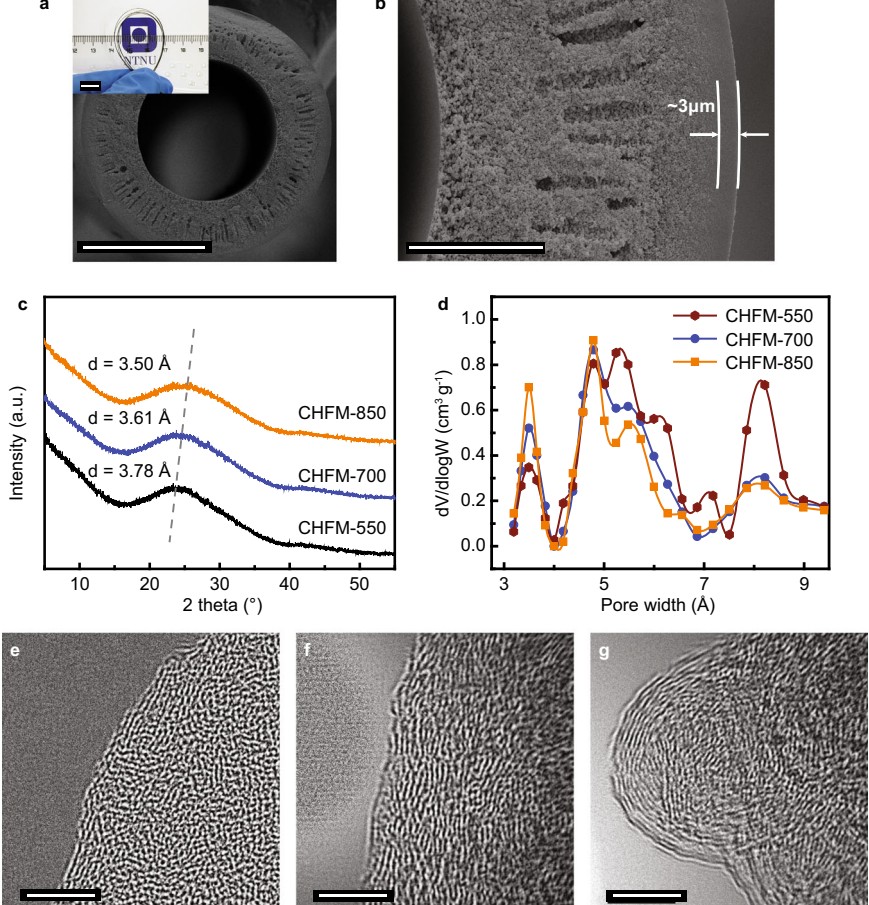

**Fig. 2 Morphology and structure characterization of the carbon hollow fiber membranes (CHFMs). a, b** Cross-sectional SEM images of CHFM-700 (carbonized at 700 °C) (scale bars: **a** 100 μm, **b** 20 μm), and the inset shows a CHFM with a bend radius of <1.5 cm (scale bar: 1 cm), indicating mechanical flexibility. The thickness of a selective layer is ca. 3 μm, and the porous morphology is well-maintained after carbonization; **c** the XRD patterns of CHFMs carbonized at different temperatures. The *d*-space was calculated from the Bragg equation, and found to decrease from 3.78 to 3.5 Å with the increase of final carbonization temperature from 550 to 850 °C, and **d** the pore size distributions of CHFMs calculated by the NLDFT model from $CO_2$ physisorption at 0 °C. The specific volumes of ultramicropores (especially the pores of <4 Å) for CHFM-850 and CHFM-700 are larger compared to that of CHFM-550, indicating more diffusion (size)-selective membranes; **e–g** HR-TEM images of CHFM-550, CHFM-700, and CHFM-850, respectively. Scale bar: 5 nm.

CHFM-850 compared to CHFM-550. Supplementary Fig. 8 compares the PSD of symmetrically dense carbon films, which is used for representing the selective layer of CHFMs. The same trend of a narrowing pore width indicates that the pore size of the selective layer, which is responsible for the molecular sieving mechanism, can be finely tuned by the carbonization temperature. Except for a similar bimodal PSD compared to carbon membranes reported in the literature[12,13,31], the present cellulose-based CHFMs exhibit a much stronger peak for the ultramicropores in the range of 3–4 Å, which is in the size range needed to allow molecular sieving between $H_2$ (2.9 Å) and other larger gas molecules (e.g., $CO_2$, $N_2$, and $CH_4$). With the increase of the carbonization temperature, the pore peaks of >5 Å are weakened, while that of the pores (<5 Å) increases, which indicates that the average pore size decreases for the CHFMs carbonized at higher temperatures. The significantly enhanced intensity of the peak for the pore width of 3–4 Å offers additional evidence for the formation of ultramicropores in CHFM-850. In addition, both the surface area and the pore volume of CHFMs (given in Supplementary Fig. 9) are reduced with the increase of the carbonization temperature, which indicates that CMS membranes tend to have more dense packing when carbonized at a higher temperature. The decreased $CO_2$ adsorption capacity of CHFM-850 (see Supplementary Fig. 10 in both the low-pressure

range of 0–1 bar and high-pressure range of 1–15 bar) shows that the higher carbonization temperature forms CMS membrane with less $CO_2$ adsorption, which is beneficial for $H_2/CO_2$ separation. High-resolution transmission electron microscopy (HR-TEM) images, as shown in Fig. 2e–g, also indicate that the CMS films are prone to be graphitized by increasing the final carbonization temperature. Although long-ordered graphitic sheets were not observed, the carbon sheets have a tendency towards a more orderly alignment over a short-range, when the carbonization temperature increased.

The hybridized carbon structure for the three CHFMs was investigated by XPS (Supplementary Fig. 12a) (and Supplementary Note 5, the XPS survey is given in Supplementary Fig. 11) and Raman spectroscopy (Supplementary Fig. 12b). The ratio of $sp^3$ to $sp^2$ hybridized carbon are calculated from the corresponding deconvoluted XPS peak areas. The ratios decrease from 0.73 to 0.36 with increasing carbonization temperature, indicating that $sp^3$-hybridized carbon transforms to $sp^2$-hybridized carbon at higher temperatures. All the CHFMs exhibit two prominent Raman peaks (Supplementary Fig. 12b), namely the G peak located at ~1600 $cm^{-1}$ corresponding to the $E_{2g}$-symmetry vibration mode of $sp^2$ hybridized carbon, and the D1 peak located at ~1346 $cm^{-1}$ corresponding to the $A_{1g}$-symmetry vibration mode from disordered graphite[32,33]. The intensities of both the G

and D1 peaks increase with carbonization temperature, which suggests a transformation towards a more graphitic carbon structure, consistent with the XPS results. But the strong D1 peak also indicates that some defects (such as the edges of the graphite[21,34]) still exist in the carbon matrix. Moreover, the slightly reduced ratio of $I_{D1}/I_G$ and the reduced intensity of the D2 peak (disordered graphitic lattice) indicate that the formed graphitic carbon tends to be more ordered at higher carbonization temperature.

During the carbonization process, the entangled polymeric precursor chains are transformed into rigid carbonized aromatic strands, and thereafter, form organized plates to approach higher entropy in the system[11,23]. A proposed mechanism for the transformation of cellulose precursors to carbon membranes is summarized in Supplementary Note 3 and Supplementary Fig. 13, which is based on characterizations of HR-TEM, XPS, Raman spectra, and thermogravimetric analysis coupled with mass spectrometry (TGA-MS) (Supplementary Fig. 14). The cellulose precursor starts depolymerization and smaller gases are evolved when the temperature is above 200 °C, and levoglucosan is formed at a temperature of 200–250 °C, resulting from cleavage of the 1,6-glycosidic linkages[35]. When the temperature was increased to ca. 400 °C, volatilized gases, like $CO_2$, $CH_4$, and $H_2O$, indicte that the levoglucosan units undergo intramolecular polymerization to form carbon plates[21]. As the temperature was increased up to 550 °C, no additional gas peaks were observed by MS. This thermal stage indicates that intramolecular rearrangement occurred to form an amorphous or less-ordered carbon structure, as shown in Fig. 2e of the HR-TEM image of CMS film carbonized at 550 °C. When the carbonization was further increased to 700 °C, small MS peaks of $CO_2$ and $H_2O$ appeared (around 600 °C), as shown in Supplementary Fig. 14b, which indicates that oxygen heteroatom evolved. As a result, a more ordered carbon structure was formed, as illustrated in Fig. 2f. Similarly, when the carbonization temperature was increased to 850 °C, both the observed peaks of $H_2O$ (at ~800 °C) and $CO_2$ (at ~850 °C) and the reduced content of oxygen detected by XPS (Supplementary Fig. 11b and Supplementary Table 3) imply that the O and H atoms were further removed, and short-range ordered graphitic sheets were formed, as depicted in Fig. 2g.

**Gas performance and transport mechanism**. The gas transport characteristics of the CHFMs were first tested using single gas permeation. The temperature-dependent $H_2/CO_2$ performance is given in Fig. 3a. Membranes prepared at higher carbonization temperatures provide higher $H_2/CO_2$ selectivity, but with the sacrifice of some $H_2$ permeance. For instance, CHFM-850 has an $H_2/CO_2$ selectivity of 46.2 at 25 °C, which is ~4 times higher than that of CHFM-550, while $H_2$ permeance is decreased from 102.1 GPU to 16.2 GPU concomitantly. In addition, there is a clear cutoff of gas permeance between the smaller molecules (148.2 GPU for $H_2$ and 139.6 GPU for He) and that of the larger molecules (Fig. 3b), which indicates that gas permeation is dominated mainly by the kinetic diameter of the gas molecules, i.e., a molecular sieving transport mechanism. The ultra-micropores are the slits or the smaller spaces between highly aromatic strands of carbon. The ultramicropores govern the gas pair selectivity, while the micropores, formed by voids between aromatic carbon plates, contribute to high gas permeance[11–13]. The cellulose-based CMS membranes reported high gas selectivities for $O_2/N_2$[20] and $CO_2/CH_4$[19]. The asymmetric cellulose-based CHFMs developed in our work present a new approach for light gas separations by creating more ultramicropores, by a combination of the coagulation temperature and non-solvent

exchange assisted morphological control during fiber spinning, as well as by tuning the carbonization temperature. As discussed previously, the CHFMs tend to form ordered graphitic carbon structures ($sp^2$ hybridized carbon) with increasing carbonization temperature. Figure 3c shows the dependence of separation performance on the $sp^3/sp^2$ ratio of different carbon membranes. When the $sp^3/sp^2$ ratio decreases from 0.73 to 0.36, the $H_2$ permeance drops markedly from 466.8 GPU to 148.2 GPU, whereas $H_2/CO_2$ selectivity increases from 11.1 to 83.9, which provides clear evidence that gas separation performance can be adjusted by tuning the carbon structure. Although the $d$-spacing of CHFM-850 (Fig. 2c) is 3.5 Å, which is larger than the $CO_2$ kinetic diameter of 3.3 Å, the achieved highest $H_2/CO_2$ selectivity of 83.9 is the result of its unique microstructure. As illustrated in Supplementary Fig. 13, the existence of oxygen-containing functional groups (such as carboxylic acid, carbonyl, and epoxy groups) between the ordered layers can strongly retard $CO_2$ transport, as reported by Kim et al.[5] Besides, the $sp^2$-hybridized carbon formed at a higher carbonization temperature provides a more ordered graphitic carbon structure, which is beneficial to the packing of the carbon strands and induces the formation of narrower ultramicropores[11]. On the other hand, the micropores existing between the aromatic carbon plates are more prone to compaction, due to the reduced content of three-dimensional $sp^3$-hybridized carbon. Gas permeance and selectivity vary significantly with temperature. Significant increases in gas permeance and selectivity are observed by increasing the temperature from 25 to 130 °C (Fig. 3a), particularly for the membranes prepared at higher carbonization temperatures. At 130 °C, the $H_2/CO_2$ selectivity and $H_2$ permeance of the CHFM-850 increased to 83.9 and 148.2 GPU, respectively, which are ~2 times and 9 times higher than the results obtained at a temperature of 25 °C. Higher temperatures accelerate gas diffusion, which enhances gas permeation. Conversely, the lower $CO_2$ adsorption at higher temperatures improves the $H_2/CO_2$ selectivity. The apparent activation energies of both the $H_2$ and $CO_2$ transport, calculated from Arrhenius plots of the gas permeance and temperature (Supplementary Note 6 and Supplementary Fig. 15), increase with temperatures. Due to the coexistence of molecular sieving and surface diffusion transport for $CO_2$ molecules, the relatively low apparent activation energy of $CO_2$ compared with $H_2$ indicates that temperature has a more significant effect on $H_2$ permeance. Therefore, considering practical industrial applications, for example, $H_2$ purification from natural gas-derived syngas (which is usually operated at 150 °C or above[3,36,37]), a higher operating temperature is preferable to enhance the $H_2/CO_2$ separation performance. The predicted higher $H_2$ permeance and $H_2/CO_2$ selectivity for CHFMs at 200 °C by extrapolating the Arrhenius regressions are also included in Fig. 3a, which can be tested in future work. The aging behavior of CHFMs was evaluated by exposing membrane modules to the laboratory atmosphere for 50 days. The $H_2$ permeance and $H_2/CO_2$ selectivity of CHFMs were reduced by about 40% and 10%, respectively (Supplementary Fig. 16), which is caused by physical and/or chemical sorption between the carbon matrix and oxygen and water molecules. A thermal treatment combined with helium sweeping through the membrane modules at a mild temperature of 100 °C for 24 h (the permeate side was operated in vacuum condition) was used to effectively recover gas permeance and selectivity. The results of Supplementary Fig. 16 indicate that 95% of the $H_2/CO_2$ selectivity can be recovered.

To test the potential of CHFMs for $H_2$ purification in a steam methane reforming process (usually performed at pressures of up to 15–20 bar), a lab-scale hollow fiber module containing CHFM-700 was tested using a 50/50 mol.% $H_2/CO_2$ mixed gas at 70 °C at different feed pressures (5–18 bar) carried out by a high-pressure

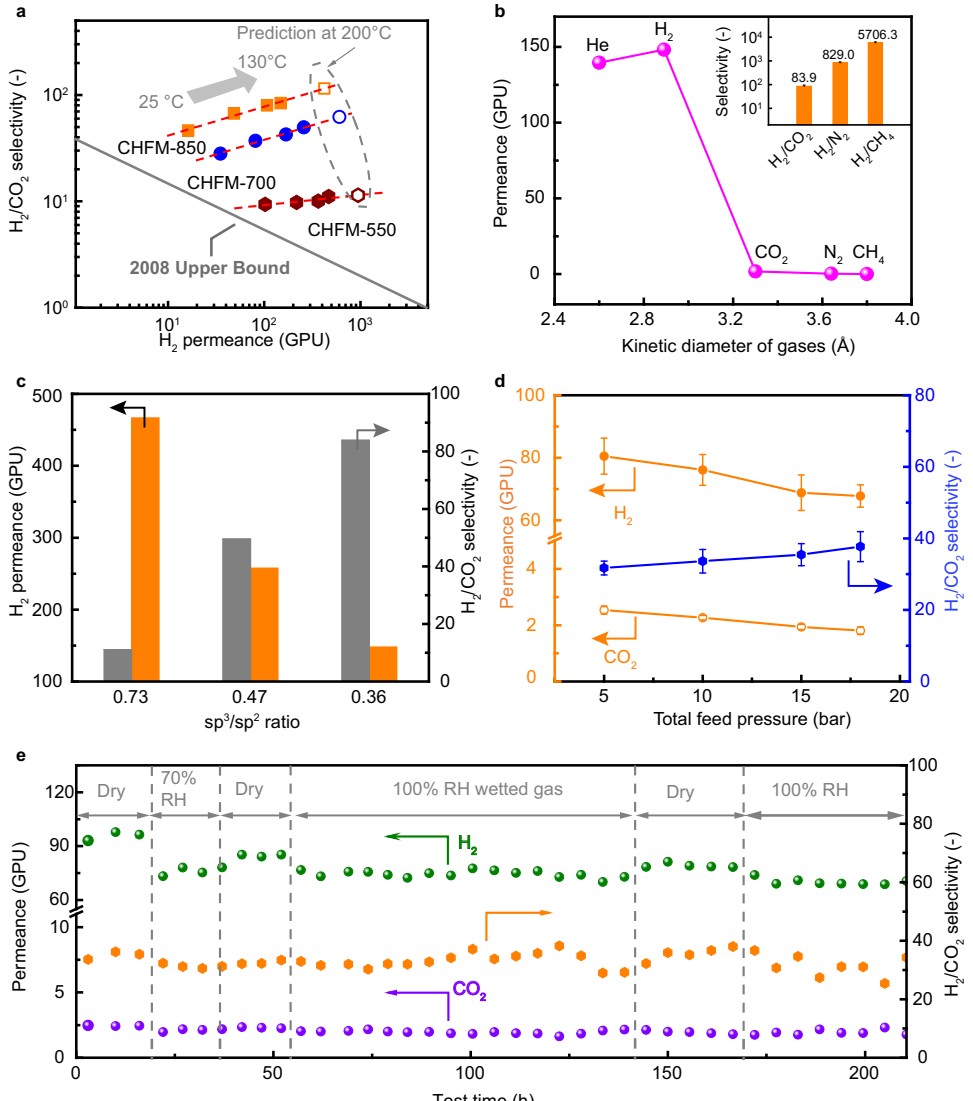

**Fig. 3 Gas separation performances of the prepared carbon hollow fiber membranes (CHFMs). a** Single-gas permeation performances of CHFMs at 25, 60, 100, and 130 °C with 2 bar feed pressure. Hollow symbols represent predicted performance at 200 °C. The gray solid line is drawn based on the 2008 Robeson upper bound by converting permeability to permeance with a thickness of 1 μm. **b** Single-gas permeances of CHFM-850 as a function of the gas kinetic diameter at 130 °C and 2 bar. The inset figure shows the selectivity of $H_2$ over $CO_2$, $N_2$, and $CH_4$ (error bars represent the standard deviation of three measurements). **c** Gas separation performance at 130 °C as a function of $sp^3/sp^2$ ratio, calculated from XPS data. **d** 50 mol% $H_2$/50 mol% $CO_2$ mixed gas measurements at different operation pressures (5–18 bar) at 70 °C (error bars represent the standard deviation of three measurements). **e** Mixed gas dynamic durability testing (50 mol% $H_2$/50 mol% $CO_2$) of CHFM-700 under dry and humidified conditions at 10 bar and 90 °C.

gas permeation rig (Supplementary Figs. 17 and 18). Figure 3d shows a gradual decrease in $H_2$ permeance (ca. 15.8%) with an increase in the total feed pressure from 5 to 18 bar, but the $H_2/CO_2$ selectivity increases from 31.8 to 37.7 (18.6% increase). $CO_2$ adsorption is gradually saturated at higher pressure (as shown in Supplementary Fig. 10b), while it is well known that the pressure influence on $H_2$ adsorption on carbon materials is much smaller, as reported by Ströbel et al.[38], especially within the investigated pressure range in our work. Therefore, the $H_2/CO_2$ selectivity slightly increases with increasing feed pressure. Here, it is important to note that conventional polymeric membranes or emerging ultrathin nanomaterials membranes (such as metal-organic frameworks, 2D nanosheets GO) either do not endure high-pressure conditions or present significantly reduced separation performance because those membrane types are more vulnerable to harsh conditions[4]. However, CMS membranes are constituted with a rigid framework that provide good pressure-

resistance. Moreover, the CHFMs prepared from cellulose hollow fiber precursors (spun from the cellulose/EmimAc system) are more easily scaled up compared with nano-2D membranes. The CHFM-700 was further assessed by dynamic durability testing over 200 h under a 50 mol% $H_2$/50 mol% $CO_2$ mixed gas with 70% RH (relative humidity) and 100% RH at 10 bar and 90 °C. As shown in Fig. 3e, the membrane was initially tested using a dry mixed gas at a pressure of 10 bar for ca. 16 h. Then the feed stream was switched to a humidified gas with 70% RH and tested over 20 h. The presence of water vapor reduced the $H_2$ permeance from 95 GPU to 75 GPU and slightly reduced the $H_2/CO_2$ selectivity from 34 to 31. When the feed stream was changed back to a dry gas mixture, the $H_2$ permeance increased to 85 GPU. The membrane module was then tested at 100% RH gas conditions over 120 h and showed stable separation performance. Such stable performance under humidified conditions can be attributed to the hydrophilic property of cellulose-derived CMS membranes. As

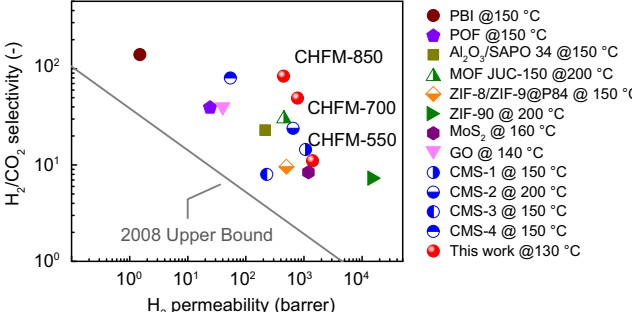

**Fig. 4 Comparison of the separation performances of the prepared carbon hollow fiber membranes (CHFMs) with state-of-the-art H₂/CO₂ separation membranes.** Comparison of gas separation performance of CHFMs developed in this work with state-of-the-art organic and inorganic membrane materials tested at 130–200 °C. The solid line is the 2008 Robeson upper bound[44]. The permeabilities of the developed CHFMs were based on a selective layer thickness of 3 μm. The details are given in Supplementary Table 4.

shown in Supplementary Fig. 19, the contact angles of water on the three types of CMS films are around 61–72°. By increasing carbonization temperatures, the contact angle slightly increases, which is the result of fewer hydrophilic –OH groups in the carbon matrix.

Figure 4 compares the $H_2/CO_2$ separation performance of the present CHFMs with other state-of-the-art polymeric (polybenzimidazole (PBI)[3,4]) and inorganic membranes[10,39–41] (ZIF[7,8], GO[5], CMS[40–43], and MoS₂[6]) in a Robeson plot[44]. The present CHFMs exhibit superior $H_2/CO_2$ selectivity and high $H_2$ permeability (assuming the thickness of the selectivity layer is 3 μm according to SEM image. A comparison of gas permeance (GPU) is given in Supplementary Fig. 20), which presents competitive performance compared with high-performance membranes reported in the literature. The CHFM-850 shows the highest overall performance, with an $H_2/CO_2$ ideal selectivity of 83.9 at 130 °C, which surpasses the non-polymeric membranes, such as ZIF, MoS₂, and GO membranes. Furthermore, to the best of our knowledge, this is the highest $H_2/CO_2$ selectivity reported for self-supported carbon membranes under the testing conditions of 130 °C and 2 bar. The separation performance of $H_2/N_2$ and $H_2/CH_4$ are given in Supplementary Fig. 21. Compared with the state-of-the-art non-polymeric membranes, the CHFM-850 still shows very attractive performance, with $H_2/N_2$ selectivity >800 and $H_2/CH_4$ selectivity >5700, which presents a great potential for $H_2$ purification in some processes (e.g., ammonia plant, $H_2$-natural gas distribution pipeline).

## Discussion

In summary, asymmetric cellulose hollow fibers were spun from microcrystalline cellulose and EmimAc, which are commercially available and do not require additional synthetic steps. The obtained cellulose hollow fibers were carbonized at three different temperatures to provide asymmetric carbon hollow fiber membranes, without the need for crosslinking for preventing pore collapse during carbonization. Due to the precise control of ultramicropore size, the present CHFMs show remarkable separation performance by molecular sieving. CHFM-850 exhibited $H_2$ permeance of 148.2 GPU and an $H_2/CO_2$ ideal selectivity of 83.9, which is the highest reported for $H_2$ purification with self-supported carbon membranes. The hydrophilic carbon membranes of CHFM-700 show good stability for over 120 h under a 100% RH humidified mixed gas at 10 bar and 90 °C, which indicates the great potential for $H_2$ purification in

steam reforming processes. The finely tuned ultramicropores of CHFMs, which result from the controlled conversion of $sp^3$ to $sp^2$ hybridized carbon by tuning carbonization temperature, provides a facile and scalable method for the preparation of high-performance CMS membranes for $H_2$ purification. The carbon membranes developed in this work are based on a renewable and low-cost polymer of cellulose, and the employed preparation process is facile, and without the requirement of additional pre-treatment to prevent pore collapse, which provides a competitive membrane production cost compared to other carbon membranes, particularly when the ILs are recovered.

## Methods

**Materials**. Microcrystalline cellulose (MCC) powder (Avicel PH-101), isopropanol (≥99.7%, FCC grade), *n*-hexane (ReagentPlus®, ≥99%), and dimethyl sulfoxide (DMSO, FCC grade) were purchased from Sigma-Aldrich. 1-Ethyl-3-methylimidazolium acetate (EmimAc, >95%) was purchased from IOLITEC GmbH. All chemicals were used as received. Single gas (e.g., $H_2$, $CO_2$) and 50 mol %-50 mol% $H_2/CO_2$ mixed gas were bought from AGA, Norway. All fittings used for the construction of membrane modules were purchased from Swagelok.

**Spinning cellulose hollow fibers**. A 12 wt.% MCC/(EmimAc+DMSO) dope solution was used in the spinning process. MCC (60 g) was gradually added into 440 g EmimAc/DMSO (weight ratio 3:1) co-solvent with mechanical stirring in an $N_2$ atmosphere glovebox, and kept at 50 °C overnight, to allow the cellulose to dissolve completely. Asymmetric cellulose hollow fibers were fabricated by a dry-wet spinning process under the conditions given in Supplementary Table 1. The resulting spun hollow fibers, which are precursors for the carbon membranes, were cut in ca 1.2 m long sections and placed in a deionized water bath over 48 h to fully exchange the solvent (EmimAc + DMSO) with water (during that time the water was replaced three times with fresh deionized water). The water-wetted cellulose hollow fibers were immersed into pure isopropanol for 2 h, followed by soaking in *n*-hexane for 2 h, and then all the hollow fibers were allowed to dry under ambient conditions in the air.

**Preparation of carbon hollow fiber membranes**. The dried cellulose hollow fiber precursor membranes were carbonized in a tubular furnace (Horizontal Split Tube Furnace, Carbolite Gero Limited) by applying the specific carbonization protocols depicted in Supplementary Fig. 6, under high purity argon (Ar, 99.999%) purge gas under a continuous flow of 80 mL min⁻¹. The tubular furnace was evacuated down to ~3 mbar overnight before being purged with Ar. A dwell-time of 2 h at 300 °C was employed to form the microstructure "carbon cell" as the chain scission and inter-molecular polymerization occur in this step (Supplementary Note 3), and a final dwell-time of 2 h at temperatures from 550–850 °C was applied to obtain ultramicropores with narrower pore size through internal condensation. The system was cooled down naturally after the carbonization process was completed, and the resulting CHFMs were removed when the temperature had cooled to below 50 °C.

The schematics of the construction of carbon membrane modules are illustrated in Supplementary Fig. 22. In brief, several CHFMs are mounted in a Swagelok 3/8-inch tubing, and both sides are sealed by epoxy adhesive (Loctite EA 3430). If membrane modules are only used for single gas permeation testing and no sweep gas is needed, one end-dead hollow fibers were made by blocking one side of fibers with epoxy adhesive, as shown in Supplementary Fig. 22b. Supplementary Fig. 23 gives the flexibility of the membrane module for singe gas testing, which is bent with a diameter of 3.7 cm. It was found that the separation performances were maintained after bending. Possible issues in membrane fabrication scale-up and potential solutions are discussed in Supplementary Note 7.

**Characterization**. SEM images were obtained using a Hitachi SU-6600 field emission scanning electron microscope (FESEM). XRD analysis of CHFMs was carried out by Bruker D8 Focus instrument operated at 45 kV and 200 mA with 2θ ranging from 5° to 70° at a scan speed of 0.05 s⁻¹ (Cu-Kα radiation, λ = 0.154 nm). $CO_2$ physisorption was measured by Quantachrome® ASiQwin™ automated gas sorption analyzer at 0 °C. High pressure $CO_2$ gravimetric sorption was conducted using a Rubotherm magnetic balance. HR-TEM images were obtained using a JEM-2100 transmission electron microscope operated at a 200 kV accelerating voltage. Gas products of cellulose pyrolysis were observed by TGA-MS (STA PT 1600, QMA 410). XPS spectra were obtained by ESCALAB 250 operated at 150 W and 200 eV with monochromatic Al-Kα radiation. Raman analysis was conducted using a Renishaw inVia Raman Microscope with a 532 nm laser. Nanoindentation tests were conducted by TriboIndenter 950 by using a Berkovich indenter (details are given in Supplementary Note 4).

**Gas permeation tests**. Single gas permeation measurements were conducted by applying a constant permeate volume method using a feed pressure of 2 bar.

 

The gas permeance and selectivity are calculated using Eq. (1):

$$\frac{P}{l} = \frac{273.15 \cdot 10^3 V}{76T \cdot A} \cdot \frac{\int_{P_i}^{P_2} \frac{dp}{P_F - p}}{\Delta t} \tag{1}$$

Where $P/l$ (GPU, 1 GPU $= 1 \times 10^{-6}$ cm$^3$(STP) cm$^{-2}$ s$^{-1}$ cmHg$^{-1} = 3.35 \times 10^{-10}$ mol s$^{-1}$ m$^{-2}$ Pa$^{-1}$) is the single gas permeance. $V$ (cm$^3$) is the downstream (permeate) volume (predetermined using He calibration), and $T$ (K) is the experimental temperature. $A$ (cm$^2$) is the hollow fiber membrane outer active surface area (shell-side feed). $P_F$ and $p$ (bar) are the pressures in the feed side and permeate side, respectively. $\Delta t$ (s) is the steady-state testing time. The H$_2$/CO$_2$ ideal selectivity is calculated by the ratio of H$_2$ permeance to CO$_2$ permeance.

The flow scheme for the 50/50 mol% H$_2$/CO$_2$ mixed gas permeation measurements is illustrated in Supplementary Fig. 17. The feed flow is controlled at 150 N mL min$^{-1}$ during the testing, and a low stage-cut of <5% was kept for all measurements. Argon was used as a sweep gas. The permeate gas flow and composition were measured by a bubble flow meter and a gas chromatograph (GC, 8610C, SRI Instruments Inc.), respectively. Three CHFM-700 membrane modules (8 carbon hollow fibers per module) were tested for dry mixed gas to determine the experimental error. One of the tested CHFM modules is shown in Supplementary Fig. 18. Gas was fed to the shell side, and the permeate gas exited from the bore side, with argon as sweep gas operated in a counter-current flow pattern. The selectivity is calculated by $\alpha = \frac{y_{H_2}/y_{CO_2}}{x_{H_2}/x_{CO_2}}$, where $y_i$ and $x_i$ are the concentration of the components in the permeate and feed, respectively.

## Data availability
The authors declare that the data supporting the findings of this study are available within the paper and its supplementary information file. Source data are provided with this paper.

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

## Acknowledgements

We thank the Research Council of Norway (Norges Forskningsråd) for funding this work through the Petromaks2 program in the CO2Hing project (#267615). Dr. Yunhan Chu is gratefully thanked for meaningful discussions. The discussions with Dr. Evangelos P. Favvas in NCSR "Demokritos" and Dr. Marius Sandru in SINTEF Industry and Prof. May-Britt Hägg are acknowledged. Mr. Sihai Luo in the Department of Chemistry, NTNU, is acknowledged for kind help with Raman analysis. Dr. Qiuxia Xu in the Institute of Process Engineering, Chinese Academy of Sciences (IPE-CAS), is acknowledged for helping with $CO_2$ physisorption measurements. We also thank Dr. Yao Li at IPE-CAS for the discussions on cellulose dissolution in ionic liquid, regeneration, and structure. Mr. Jianyu Ma and Feng Wang at NTNU are also acknowledged for helping on TGA-MS and mechanical strength analysis, respectively.

## Author contributions

L.L., A.L., and X.H. conceived the research and carried out the experiment. M.D.G. provided suggestions for the structure of the results and discussion. L.L. performed the pure and mixed gas permeation test, SEM characterization. L.L, F.P., and L.B. performed XRD, HR-TEM, XPS, and $CO_2$ physisorption analysis. X.Z., Y.N., and M.H. helped to revise the paper. L.L., X.H., and M.D.G. wrote the paper. All authors participated in the discussion and approved the final version of the manuscript.

## Competing interests

L.L, A.L., and X.H. hold a filed patent application related to the development of the carbon hollow fiber membranes. The remaining authors declare no competing interests.
