## [Peer Review File · Nature Communications]

REVIEWER COMMENTS

Reviewer #1 (Remarks to the Author):

Title: Carbon Molecular Sieve Hollow Fiber Membranes with Precise-cutoff Ultramicropores for Superior Hydrogen Separation

In this study, authors proposed the facile method for fabricating cellulose-based asymmetric carbon hollow fiber membrane by adjusting the coagulation temperature to form the asymmetric polymeric hollow fiber. Subsequently, the cellulose-derived carbon hollow fiber membranes with ultramicropores of 3-4 angstrom were prepared by tuning different carbonization temperature to exhibits unprecedented H₂/CO₂ selectivity of 83.9 and hydrogen permeance of 148.2 GPU at permeation temperature of 130 °C. Based on the performance comparison with other inorganic membranes and polymeric membranes, the permselectivity of asymmetric carbon hollow fiber membrane indeed outperforms all membranes and provides the promising candidate for gas separation field. The novelties of this work stated by authors lie in the configuration of cellulose-based membrane (hollow fiber) and the superior H₂/CO₂ selectivity.

The results also showed the good mechanical strength, and high-pressure resistance which has the potential could be applied in methanol steam reforming. Referee definitely believes that the CHFM synthesized in this work showing the remarkable separation performance would attract much attention to gas separation membrane-related communities. However, the following comments still need to be addressed before considering to be accepted by Nature Communication.

1. There are the published papers elsewhere for cellulose-derived CHFM [1, 2], as follows. Besides, it is well known that the performance of cellulose-derived carbon molecular membrane is excellent for hydrogen separation over H₂/N₂ and H₂/CH₄. Therefore, the authors should clarify the difference in the mechanism or interaction of as-obtained carbon hollow fiber membrane over the overwhelming H₂/CO₂ selectivity, in comparison to other carbon membranes prepared from the same precursor (cellulose) or other thermosetting polymer precursor, e.g., polyimide, polyetherimide and phenolic resin. [1] Preparation and Characterization of Hollow Fiber Carbon Membranes from Cellulose Acetate Precursors, *Industrial & Engineering Chemistry Research*, 50, 2011, 2080-2087. [2] Pilot-Scale Production of Carbon Hollow Fiber Membranes from Regenerated Cellulose Precursor-Part II: Carbonization Procedure, *Membranes (Basel)*, 8, 2018.
2. The pore size distribution of carbon hollow fiber membrane is analyzed by CO₂ physisorption technique. However, the analysis limitation of this technique is 3.2 angstrom. How come the pore size (lower than 3.2 angstrom) of obtained carbon hollow fiber membrane could be detected? As shown in Fig. 2(d), the Dv/dlogW of ultramicropores of CHFM were increased with the carbonization temperature increased. On this basis, I do not understand what Fig 2(d) actually represents or how representative it really is of the bulk of the CMS-or how it relates to the properties of the CMS materials that the authors have made. These issues MUST be clarified in a revised manuscript before the work is seriously considered for publication.
3. The above point is important, since I am unclear exactly what morphology the current authors believe exists in a cellulose-derived CMS based on Fig. S8 and the associated discussion of that figure. They note on page 3 that CMS has a bimodal pore size distribution, but it is highly unlikely that significant graphitic sheets the a pyrolyzed sample to allow the performance they report.
4. Although the cut-off pore size can be adjusted by vary the carbonization temperature, it was strongly dependent on the chemical structure of cellulose precursor used in this study. There is no discussion on the relationship between them.
5. As shown in Fig. 3, the gases permeabilities obtained from the CHFM were followed the sequence of the kinetic diameters of gases, indicating that the transport mechanism was dominated by molecular sieving mechanism, an activation process. Thus, the activation energy of CO₂ gas is expected to be higher than that of H₂ (Fig. S11).
6. Because of the water-wetted property of cellulose membrane, the aging phenomenon of derived carbon membrane should be investigated, if it will be applied in steam reforming reaction.
7. As shown in the gas permeation results of CHFM-850, the permeance of helium is lower than that of

hydrogen, which is contradictory to molecular sieving mechanism.

8. As shown in Raman spectra of obtained carbon hollow fiber membrane as function of pyrolysis temperature, it clearly observed the intensity of D1 peak arisen from CHFM-850 maintained in comparison to other membranes at lower pyrolysis temperature, implying the packing defects still exist. However, the ratio of sp³/sp² decreased with rising the carbonization temperature as shown in XPS results, indicating the structure of carbon membrane become more ordered.

9. The authors claimed the discrimination in size between ultra-micropore and micropore is 5 angstroms. However, according to the envisioned structure of carbon molecular sieving membrane as reported by William J. Koros et al., the size of micropores is ranging from 7-20 angstrom, while that of ultra-micropore is lower than 7 angstrom. I concern whether the difference in judging the kind of
10. The paper should be carefully checked and several missing unit of temperature (°C) should be added.

(1) Grammatical errors at Page No. 6, Line No. 117: "...noting..." should be "noted"

(2) Missing units at the title of Fig. 1a

(3) Missing units at Page No. 14, Line No. 306

(4) Missing units at Line No. 104 in supplemental file

Reviewer #2 (Remarks to the Author):

As the authors note, asymmetric cellulose-based carbon hollow fiber membranes have not been reported for gas separation. As the authors also note, solubility limitation require the use of rather exotic solvents such a as N-methylmorpholine-N-oxide (NMMO), ionic liquids and inorganic salts to achieve dopes here to allow flat sheet and hollow fibers here. This report extends their earlier work reported in I&ECR, 2019, 58, 13330-13339 paper (ref. 17) on screening of spinning parameters and applied primarily to the CO₂/CH₄ separation. While successful in terms of CO₂/CH₄ selectivity, the acheivable permeability and permeance (permeability divided by selective layer thickness) were very low, and essentially impractical for that application.

A real advancement reported here that goes beyond the I&ECR paper is that the authors have now carefully optimized the quench bath temperature as a new variable to allow achieving asymmetry in the precursor. This additional variable allows maintaining some asymmetry in the CMS.

The CO₂ permeance, while higher than in the I&ECR paper is still quite low, but they now focus on H₂/CO₂ for hydrogen purification from the steam methane reforming process, rather than CO₂/CH₄ separation. In their paper they also note the ability to avoid treatments required for other precursors to avoid collapse for the morphology during pyrolysis.

I feel that the work is quite interesting but incomplete for such an application, since the steam-methane reforming process requires stability of the membrane to do the H₂/CO₂ separation in the presence of high temperatures and steam. If the authors will complement their current results with at least some results for ternary feeds comprising not only H₂ & CO₂, but also steam at temperatures up to 130 °C, I think the work would meet the standards of Nature Communications. On this basis, I note "major revisions" are required for the authors to accommodate this important issue. While it might be acceptable for the authors to only discuss and justify why they think the ternary results are not required, I think this is highly questionable and should be addressed with actual data.

Reviewer #3 (Remarks to the Author):

The paper does not report a striking novelty – but the healthy development of a practice-relevant

carbon membrane with a chance of commercialization. By a clever selection of the starting materials and sophisticated carbonization conditions, self-supporting carbon hollow fiber membranes with asymmetric cross sections have been developed. The characterization by XPS (sp^2 - sp^3) and Raman is correct, but not new. After balancing the different opinions, I prefer a publication of the manuscript with a 60-40 opinion. Since the manuscript is well prepared and the techniques are well described, there is not much space at the bottom and here are only a few remarks for a revision.

- Fibers can be bent with a radius of 1.5 cm. Permeation after bending? there further mechanical properties (Young module) of the self-supporting hollow fibers?
- It is common that membrane people use Robeson's upper bound from 2008 as a benchmark like a holy grail. But this was 12 years ago. Are there new developments in the world of polymer membranes in the last 12 years?
- Authors say that the carbon membranes are cheap using cellulose. But also an IL is used. The price of the IL depends on its purity and is between 300 and 1000 €/kg. Is this a problem?
- Some graphics need improvements:
Fig. 3a: H₂/CO₂ Selectivity contra Fig. 4: H₂/CO₂ selectivity
Fig. 3a: H₂ Permeance contra Fig. 4: H₂ permeability
- Apply Dunitz' rule: The first sentence of every manuscript can be deleted, e.g. "Fossil fuels are limited / Cancer is a plague / Global warming is a threat to humanity"
- Can the authors give some application-relevant remarks: How can these fibers be fixed in a module? Regeneration? Scale up of the fiber fabrication?

RE: Point-by-point response for revision to manuscript NCOMMS-20-18017

Title: Carbon Molecular Sieve Hollow Fiber Membranes with Precise-cutoff Ultramicropores for Superior Hydrogen Separation

Authors: Linfeng Lei, Fengjiao Pan, Arne Lindbråthen, Xiangping Zhang, Magne Hillestad, Yi Nie, **Lu Bai**, Xuezhong He, Michael D. Guiver

August 23rd, 2020

Response to Reviewers' comments:

The authors would like to thank the Reviewers and the Editor for their constructive comments and suggestions to help us improve our research and the quality of this manuscript. We have carefully considered all the Reviewers' comments and have revised the manuscript to address their concerns. To aid in the reviewing process, we have replied to all the comments on a point-by-point basis and highlighted the revised sections of the main manuscript and Supplementary Information with red font color. We hope the manuscript can now be accepted for publishing in *Nature Communications*.

The authors confirm we now include Dr. Lu Bai from the Institute of Process Engineering, Chinese Academy of Sciences in the author list for her contribution to structure characterization by HR-TEM and CO₂ sorption analysis of carbon film samples.

On behalf of the authors, and with kind regards,
Dr. Michael D. Guiver (designated corresponding author for the submission)
Dr. Xuezhong He (co-corresponding author)

REVIEWER COMMENTS

Reviewer #1 (Remarks to the Author):

Title: Carbon Molecular Sieve Hollow Fiber Membranes with Precise-cutoff Ultramicropores for Superior Hydrogen Separation

In this study, authors proposed the facile method for fabricating cellulose-based asymmetric carbon hollow fiber membrane by adjusting the coagulation temperature to form the asymmetric polymeric hollow fiber. Subsequently, the cellulose-derived carbon hollow fiber membranes with ultramicropores of 3-4 angstrom were prepared by tuning different carbonization temperature to exhibits unprecedented H₂/CO₂ selectivity of 83.9 and hydrogen permeance of 148.2 GPU at permeation temperature of 130 °C. Based on the performance comparison with other inorganic membranes and polymeric membranes, the permselectivity of asymmetric carbon hollow fiber membrane indeed outperforms all membranes and provides the promising candidate for gas separation field. The novelties of this work stated by authors lie in the configuration of cellulose-based membrane (hollow fiber) and the superior H₂/CO₂ selectivity.

The results also showed the good mechanical strength, and high-pressure resistance which has the potential could be applied in methanol steam reforming. Referee definitely believes that the CHFM synthesized in this work showing the remarkable separation performance would attract much attention to gas separation membrane-related communities. However, the following comments still need to be addressed before considering to be accepted by Nature Communication.

Author Answer: We highly appreciate the reviewer's positive comments. We reply to your comments below:

Reviewer R1-1. There are the published papers elsewhere for cellulose-derived CHFM [1, 2], as follows. Besides, it is well known that the performance of cellulose-derived carbon molecular

membrane is excellent for hydrogen separation over H_2/N_2 and H_2/CH_4 . Therefore, the authors should clarify the difference in the mechanism or interaction of as-obtained carbon hollow fiber membrane over the overwhelming H_2/CO_2 selectivity, in comparison to other carbon membranes prepared from the same precursor (cellulose) or other thermosetting polymer precursor, e.g., polyimide, polyetherimide and phenolic resin. [1] Preparation and Characterization of Hollow Fiber Carbon Membranes from Cellulose Acetate Precursors, *Industrial & Engineering Chemistry Research*, 50, 2011, 2080-2087. [2] Pilot-Scale Production of Carbon Hollow Fiber Membranes from Regenerated Cellulose Precursor-Part II: Carbonization Procedure, *Membranes (Basel)*, 8, 2018.

Author Answer A1-1: Our previously reported cellulosic-based carbon membranes in these two references (using deacetylated cellulose acetate, DCA) did present relatively good separation performance for H_2/N_2 , and H_2/CH_4 . However, the DCA-derived carbon membranes present symmetric structure after the deacetylation treatment and thus have low gas permeance. There are two challenges to produce cellulose membranes from DCA: a required additional step of deacetylation, and a subsequent drying step. As reported in refs. [1] and [2], an average yield of useful hollow fibers is about 80% (for some batches, even 20% yield is observed). Considering large-scale production, a 20% product loss of hollow fibers before carbonization is high. This is one of the reasons why the authors are trying to develop asymmetric cellulose hollow fiber membranes with high gas permeance thin selective layers directly from cellulose as a raw material.

Compared with other polymeric precursors, like polyimides, current cellulose-based CMS membranes show high gas selectivities but relatively low permeances [3-4]. As the reviewer suggested in comment 4, the separation performances are also dependent on the chemical structures of precursors. Heteroatoms like N and F present in polymer chains significantly affect the carbon matrix during the carbonization process. For example, recent work from the group of Prof. Koros [5] reported carbonized polyimide wherein the atom content of nitrogen in the final carbon matrix is about 3%. The existence of N at the sp^3 carbon edge and the sp^2 carbon-ring plane might exert a significant influence on the fine carbon structure. Furthermore, it has also been reported that doping heteroatoms, such as N and F, into carbon materials could enhance CO_2 -philic properties and hence result in high CO_2 selectivity materials [6,7]. For the carbon membranes developed in this work, the cellulose precursor only contains C, H, O atoms, which provide the possibility of fine-tuning the carbon structure. It should be noted that the very low N content of <0.6% detected by XPS likely arises from the residual solvents for ionic liquids, which should not influence the sp^2 carbon structure (i.e., carbon planar layer), since the N contents in the most of the reported N-doped carbon materials are >3%.

Reference:

- [1] Haider S, Lie JA, Lindbråthen A, Hägg M-B. Pilot-Scale Production of Carbon Hollow Fiber Membranes from Regenerated Cellulose Precursor-Part I: Optimal Conditions for Precursor Preparation. *Membranes* 8, 105 (2018).
- [2] Haider S, Lie JA, Lindbråthen A, Hägg M-B. Pilot-Scale Production of Carbon Hollow Fiber Membranes from Regenerated Cellulose Precursor-Part II: Carbonization Procedure, *Membranes* 8 97 (2018).
- [3] Lei L, Lindbråthen A, Hillestad M, Sandru M, Favvas EP, He X. Screening Cellulose Spinning Parameters for Fabrication of Novel Carbon Hollow Fiber Membranes for Gas Separation. *Industrial & Engineering Chemistry Research* 58, 13330-13339 (2019).
- [4] Rodrigues SC, Whitley R, Mendes A. Preparation and characterization of carbon molecular sieve membranes based on resorcinol-formaldehyde resin. *Journal of Membrane Science* 459, 207-216 (2014).
- [5] Qiu W, et al. Hyperaging Tuning of a Carbon Molecular-Sieve Hollow Fiber Membrane with Extraordinary Gas-Separation Performance and Stability. *Angewandte Chemie International Edition* 58, 11700-11703 (2019).
- [6] To JWF, et al. Hierarchical N-Doped Carbon as CO_2 Adsorbent with High CO_2 Selectivity from Rationally Designed Polypyrrole Precursor. *Journal of the American Chemical Society* 138, 1001-1009 (2016).
- [7] Yang Z, et al. Surpassing Robeson Upper Limit for CO_2/N_2 Separation with Fluorinated Carbon Molecular Sieve Membranes. *Chem* 6, 631-645 (2020).

Action: The influence of heteroatoms doped in CMS materials has been included in the revised manuscript:

P4: ...“It has been reported that doping heteroatoms, such as N and F, into carbon materials could enhance the CO₂-philic properties and hence result in highly CO₂-selective materials^{17,18}. To et al.¹⁷ have demonstrated Henry’s Law selectivity of CO₂/N₂ for N-doped porous carbons was increased from 9 to 124 when the N content was increased from 3.2 to 5.8 wt.%. Similarly, Yang et al.¹⁸ reported that the diffusion of CO₂ molecules through the membrane can be improved by the O-, N- and F-containing nanodomains. On the contrary, designing a low heteroatom-content CMS material toward limiting CO₂ affinity is a feasible approach to achieve high H₂/CO₂ selective carbon membranes.”...

R1-2. The pore size distribution of carbon hollow fiber membrane is analyzed by CO₂ physisorption technique. However, the analysis limitation of this technique is 3.2 angstrom. How come the pore size (lower than 3.2 angstrom) of obtained carbon hollow fiber membrane could be detected? As shown in Fig. 2(d), the Dv/dlogW of ultramicropores of CHFMs were increased with the carbonization temperature increased. On this basis, I do not understand what Fig 2(d) actually represents or how representative it really is of the bulk of the CMS-or how it relates to the properties of the CMS materials that the authors have made. These issues MUST be clarified in a revised manuscript before the work is seriously considered for publication.

A1-2: Thank you for the comment. The pore size distribution (PSD) was automatically obtained from the NLDFT model analysis provided by the instrument software (Quantachrome® ASiQwin™). The recent work (*Carbon* **168**, 508-514 (2020)) compared different methods, like NLDFT (Quantachrome® ASiQwin™, NLDFT (SAIEUS Micromeritics) and Grand Canonical Monte Carlo simulations (GCMC), and reported that the PSD in the ultramicroporous region (width < 0.7 nm) were very similar. Thus, the authors believe that the method used in this work should be correct. As shown in the original manuscript **Fig. 2d**, the first point starts at 3 Å with a Dv/dlogW = 0, while the second point starts at 3.2 Å. The authors agree that <3.2 Å pore sizes are outside the detection range, and the data range from 3-3.2 Å has no physical meaning, and thus we have now removed the first sorption data point at 3 Å in the revised manuscript (see **Fig. R1**). It should be noted that this does not affect the trends of the PSD change with increasing carbonization temperature.

Fig. R1. Pore size distribution of a) CHFMs calculated by the NLDFT model from CO₂ physisorption at 0 °C.

The authors believe that the PSD of the porous layer of CHFMs cannot provide a significant contribution to molecular sieving separation of H₂/CO₂ as it is too porous based on the SEM images. In order to investigate the microstructure of the selective layer, instead of the bulk CMS membranes, flat-sheet cellulose films with different thicknesses were cast and dried under the same condition as we have discussed in original Supplementary Information. In brief, 25 °C dope solution was cast in different thicknesses and coagulated in a 60 °C water bath. The wetted films were then exposed to solvent exchange using isopropanol and n-hexane. As shown in **Fig. R2**, the selective layer thickness of the prepared cellulose films is about 9-10 μm while the porous layer varies. When a thin film of less than 10 μm is made (**Fig. R2a**), the bulk film has entirely dense morphology, which is representative of the selective layer of our CHFMs. Thus, the thinnest dense films (**Fig. R2a**) were carbonized, and the obtained carbon films were further used to characterize the structure and PSD by TEM and CO₂ sorption analysis.

Fig. R2. Comparative cross-sectional SEM images of flat-sheet membranes cast with various thicknesses. a) the whole thickness is the dense selective layer; b) and c) asymmetric films with dense selective layer and porous support layer. Scale bars: 20 μm.

As shown in **Fig. R3**, The PSD patterns of the dense CMS films show similar distributions with the PSD of the CHFMs shown in **Fig. R1**, which provides supporting evidence that the PSD of the selective layer of CMS materials can be tuned by increasing the carbonization temperature to enhance the molecular sieving separation performance.

Fig. R3. Pore size distribution of carbon films, calculated by the NLDFT model from CO₂ physisorption at 0 °C.

Action: **Fig. 2d** was revised by removing the first data point of 3 Å. Flat-sheet cellulose films with various thicknesses were prepared from casting solutions using the same coagulation temperature of the spinning process. The cross-sectional SEM images of the prepared films have been added in the revised Supplementary Information (**Supplementary Fig. 3**). The symmetric dense thin films (**Fig. R2a**) were carbonized at temperatures of 550 °C, 700 °C, and 850 °C, and the resulting carbon films were characterized by CO₂ physisorption to obtain their PSDs, which has now been added in the

Supplementary Information (**Supplementary Fig. 8**). Additional detailed discussions on PSD have also been included in the revised manuscript and Supplementary Information:

In the revised manuscript:

P8: ...“**Supplementary Fig. 8** compares the PSD of symmetrically dense carbon films, which is used for representing the selective layer of CHFMs. The same trend of a narrowing pore width indicates that the pore size of the selective layer, which is responsible for the molecular sieving mechanism, can be finely tuned by the carbonization temperature.”...

In the revised Supplementary Information:

P1: ...“Flat-sheet cellulose films with different thicknesses were cast and dried under the same conditions as those used for the spinning process, as shown in **Supplementary Fig. 3**. When a thin film with a thickness of smaller than 10 μm was made (**Supplementary Fig. 3a**), the bulk film presented entirely dense morphology, which is representative of the selective layer of the CHFMs. Thus, the thinnest cellulose films were carbonized, and the obtained carbon films were used for structural characterization.”...

R1-3. The above point is important, since I am unclear exactly what morphology the current authors believe exists in a cellulose-derived CMS based on Fig. S8 and the associated discussion of that figure. They note on page 3 that CMS has a bimodal pore size distribution, but it is highly unlikely that significant graphitic sheets in a pyrolyzed sample allow the performance they report.

R1-4. Although the cut-off pore size can be adjusted by varying the carbonization temperature, it was strongly dependent on the chemical structure of cellulose precursor used in this study. There is no discussion on the relationship between them.

A1-3 and A1-4: Thanks for the good questions. As these two comments are both related to the structure evolution from cellulose precursors to CMS membranes, which are replied here collectively. The authors agree that the separation performance is dependent on the structure of CMS membranes and the chemical structure of cellulose precursor. Firstly, CMS membranes with a bimodal pore size distribution of micropores and ultramicropores have been widely reported, such as, by Koh et al. (*Science* **353**, 804 (2016)), Zhang and Koros (*Advanced Materials* **29**, 1701631 (2017)), and Rungta, et al. (*Carbon* **115**, 237-248 (2017)). In this work, the CO_2 physisorption data also shows the prepared CMS membranes have a bimodal PSD. Thus, it seems reasonable to draw the assumption that the CMS membranes made in this work present a bimodal PSD. It should be noted that the prepared membranes do not have a fully graphite-like carbon structure based on the TEM, XRD and XPS analysis (even though CHFMs-850 has short-range ordered sheets (see Fig. **R4** below) and its d-space is close to 3.4 Å for graphite). The authors agree that the illustration of **Fig. S8** with a highly graphitic structure may not fully represent the actual carbon membrane structure obtained in this work, and thus this has now been modified in the revised Supplementary Information.

To examine the fine microstructure of the CMS membranes, CMS thin films derived from the cellulose films with a thickness of $\sim 9 \mu\text{m}$ (**Fig. R2a**) were used to represent the selective layer of the reported CHFMs and characterized by high-resolution transmission electron microscopy (HR-TEM). As shown in **Fig. R4**, the CMS films are prone to be graphitized by increasing the final carbonization temperature. Although long-ordered graphitic sheets were not observed, the carbon sheets have the tendency to be aligned more orderly over a short range, when the carbonization temperature increased. Besides, the thermogravimetric analysis coupled with mass spectrometry (TGA-MS) was also employed to investigate the evolved small gas molecules during carbonization. As illustrated in **Fig. R5**, four main gas molecules- CO_2 , H_2 , H_2O and CH_4 - were detected by MS at different carbonization procedures. It is clearly shown that the cellulose precursor starts depolymerization and smaller gases are formed when the temperature is above 200 °C. It was also suggested that levoglucosan was formed at a temperature of 200-250 °C, resulting from cleavage of the 1,6-glycosidic linkages, based on a ReaxFF molecular dynamic simulation (*Fuel* **177**, 130-141 (2016)). When the temperature was increased to ca. 400 °C, volatilized gases, like CO_2 , CH_4 and H_2O , induced the levoglucosan units to undergo intramolecular polymerization to form carbon plates (*Carbon* **150**, 142-152 (2019)). As the

temperature was increased to 550 °C, no additional gas peaks were observed by MS. This thermal stage indicates that intramolecular rearrangement occurred to form an amorphous or less-ordered carbon structure, as shown in **Fig. R4a** of the HRTEM image of CMS film carbonization at 550 °C. When the carbonization was further increased to 700 °C, small MS peaks of CO₂ and H₂O were detected (around 600 °C), as shown in **Fig. R5b**, which indicates that oxygen heteroatom evolved. As a result, a more ordered carbon structure was formed, as illustrated in **Fig. R4b**. Similarly, when the carbonization temperature was increased to 850 °C, the observed peaks of H₂O (at ~ 800 °C) and CO₂ (at ~ 850 °C) implies that the O and H atoms were further removed, and short-range ordered graphitic sheets were formed, as depicted in **Fig. R4c**.

Based on the characterization results of HR-TEM, XPS, Raman spectra and TGA-MS, we have revised the transformation mechanism from cellulose precursors to CMS membranes, as is proposed in **Fig. R6**. When the final carbonization temperature is below 600 °C (in this work, 550 °C was used), disordered carbon “plates” were formed by intramolecular rearrangement, and a higher content of oxygen heteroatom existed in the carbon matrix by the formation of -OH, -COO, and -CH₃ groups, which contributed to the more disordered structure (**Fig. R6c**). As the final carbonization temperature was increased to over 600 °C, and especially over 800 °C, pendant groups, such as -OH and -CH₃, were removed by forming H₂O and CO₂, which resulted in a more ordered carbon structure (**Fig. R6d**). This was also supported by the higher sp² carbon content and lower oxygen content in the XPS spectra. Furthermore, according to HR-TEM and PSD, it can be proposed that the ultramicropores are from the inter-planar spacing, while the micropore contribution is from the imperfect packing of the carbon sheets.

Fig. R4. HR-TEM images of thin CMS films prepared from the carbonization temperature of a) 550 °C, b) 700 °C, and c) 850 °C. scale bar: 5 nm.

Fig. R5. The evolved small gas molecules CO_2 , H_2 , H_2O , and CH_4 from cellulose carbonization under different carbonization procedures as measured by TGA-MS.

Fig. R6. A proposed mechanism for the transformation of cellulose precursors to CMS membranes.

Action: Characterization using HR-TEM and TGA-MS was conducted to investigate the fine structure of CMS membranes and the structure evolution of CMS membranes from cellulose precursor. The HR-TEM images have been included in the revised manuscript (Fig. 2e-g) and Fig. R5 and R6 were included in the Supplementary Information (Supplementary Fig. 14 and Fig. 13). More discussion has been included in the revised manuscript and Supplementary Information.

In the revised manuscript, the following paragraphs have now been included:

P8-9: ...“High-resolution transmission electron microscopy (HR-TEM) images, as shown in **Fig. 2e-g**, also indicate that the CMS films are prone to be graphitized by increasing the final carbonization temperature. Although long-ordered graphitic sheets were not observed, the carbon sheets have a tendency towards a more orderly alignment over a short range, when the carbonization temperature increased.”...

P9-10: ...“ A proposed mechanism for the transformation of cellulose precursors to carbon membranes is summarized in **Supplementary Note 3** and **Supplementary Fig. 13**, which is based on characterizations of HR-TEM, XPS, Raman spectra, and thermogravimetric analysis coupled with mass spectrometry (TGA-MS) (**Supplementary Fig. 14**). The cellulose precursor starts depolymerization and smaller gases are formed when the temperature is above 200 °C, and levoglucosan was formed at a temperature of 200-250 °C, resulting from cleavage of the 1,6-glycosidic linkages³⁵. When the temperature was increased to ca. 400 °C, volatilized gases, like CO₂, CH₄, and H₂O, induced the levoglucosan units to undergo intramolecular polymerization to form carbon plates²¹. As the temperature was increased up 550 °C, no additional gas peaks were observed by MS. This thermal stage indicates that intramolecular rearrangement occurred to form an amorphous or less-ordered carbon structure, as shown in **Fig. 2e** of the HRTEM image of CMS film carbonization at 550 °C. When the carbonization was further increased to 700 °C, small MS peaks of CO₂ and H₂O (around 600 °C), as shown in **Supplementary Fig. 14b**, which indicates that oxygen heteroatom evolved. As a result, a more ordered carbon structure was formed, as illustrated in **Fig. 2f**. Similarly, when the carbonization temperature was increased to 850 °C, the observed peaks of H₂O were detected (at ~ 800 °C) and CO₂ (at ~ 850 °C) implies that the O and H atoms were further removed, and short-range ordered graphitic sheets were formed, as depicted in **Fig. 2g**.”...

P17: ...“HRTEM images were obtained using a JEM-2100 transmission electron microscope operated at a 200 kV accelerating voltage. Gas products of cellulose pyrolysis were observed by TGA-MS (STA PT 1600, QMA 410).”...

In the revised Supplementary Information:

P3:...“**Supplementary Fig. 13** outlines the transformation mechanism from cellulose precursors to CMS membranes, based on the characterization results of HR-TEM, XPS, Raman spectra and TGA-MS. When the final carbonization temperature is below 600 °C (in this work, 550 °C was used), disordered carbon “plates” were formed by intramolecular rearrangement, and a higher content of oxygen heteroatom existed in the carbon matrix by the formation of –OH, –COO, and –CH₃ groups, which contributed to the more disordered structure (**Supplementary Fig. 13c**). As the final carbonization temperature was increased to over 600 °C, and especially over 800 °C, pendant groups, such as –OH and –CH₃, were removed by forming H₂O and CO₂, which resulted in a more ordered carbon structure (**Supplementary Fig. 13d**). This was also supported by the higher sp² carbon content and lower oxygen content in the XPS spectra. Furthermore, according to HR-TEM and PSD, it can be proposed that the ultramicropores are from the inter-planar spacing, while the micropore contribution is from the imperfect packing of the carbon sheets.”...

R1-5. As shown in Fig. 3, the gases permeabilities obtained from the CHFMs were followed the sequence of the kinetic diameters of gases, indicating that the transport mechanism was dominated by molecular sieving mechanism, an activation process. Thus, the activation energy of CO₂ gas is expected to be higher than that of H₂ (Fig. S11).

A1-5: For CMS membranes, both molecular sieving and surface diffusion transport mechanisms exist. For selective surface diffusion, the permeate gases show a strong affinity with the CMS membrane surface and are adsorbed along the pore wall, which allows CO₂ molecules to pass through the membrane easily. If a CMS membrane is dominated by selective diffusion, a CO₂/H₂ selective

membrane can be obtained, as reported by Richter et al. (*Angewandte Chemie International Edition* **56**, 7760-7763 (2017)). Due to the coexistence of molecular sieving and surface diffusion transport mechanisms for CO₂ gas, a low activation energy was obtained, which was calculated from the Arrhenius relationship of permeability and temperature. In the case of H₂ molecules, gas permeation is dominated by diffusivity, with a relatively low contribution from sorption. Overall, H₂ presents a higher activation energy compared to CO₂ for our carbon membranes.

Action: The explanation of the higher activation energy of H₂ than that of CO₂ has been included in the revised manuscript.

P12: ...“Due to the coexistence of both molecular sieving transport and surface diffusion transport for CO₂ molecules, the relatively low apparent activation energy of CO₂ compared with H₂ indicates that temperature has a more significant effect on H₂ permeance.”...

R1-6. Because of the water-wetted property of cellulose membrane, the aging phenomenon of derived carbon membrane should be investigated, if it will be applied in steam reforming reaction.

A1-6: Thanks for the suggestion. After the CHFMs were exposed in the lab atmosphere for 5 days and 50 days, the single gas performances were measured to characterize the membrane aging phenomena (for the water-contained gas permeation experiments (i.e., the simulated steam methane reforming conditions), please refer to the answer **A2-1**). As summarized in **Fig. R7**, after being exposed to lab atmosphere for 50 days, the H₂ permeance and H₂/CO₂ selectivity of CHFMs was reduced by about 40% and 10%, respectively. This is caused by physical and/or chemical sorption between carbon matrix and oxygen and water molecules. For recovering the performance, a thermal treatment was employed for performance regeneration. Here, the feed side of the membrane modules was filled with helium at a temperature of 100 °C for 24 h while the permeate side was evacuated under vacuum. As shown in **Fig. R7**, the separation performances of CHFMs recovered slightly. Since the thermal regeneration treatment used in this work was conducted at a relatively mild temperature, the adsorbed impurities might not be removed completely. Thus, the loss of performances was not fully recovery. Other regeneration methods will be considered in future work.

Fig. R7. Normalized H₂ permeance (a) and H₂/CO₂ selectivity (b) as a function of aging time, tested at 130 °C for single gas.

Action: **Fig. R7** was added in the supplementary information (**Supplementary Fig. S15**). Aging phenomenon has now been investigated and has been included in the revised manuscript:

P12: ...“ The aging behavior of CHFMs was evaluated by exposing membrane modules to the laboratory atmosphere for 50 days. The H₂ permeance and H₂/CO₂ selectivity of CHFMs

were reduced by about 40% and 10%, respectively (**Supplementary Fig. 16**), which is caused by physical and/or chemical sorption between the carbon matrix and oxygen and water molecules. A thermal treatment combined with helium sweeping through the membrane modules at a mild temperature of 100 °C for 24 h (the permeate side was operated under vacuum) was used to effectively recover gas permeance and selectivity. The results of **Supplementary Fig. 16** indicate that 95 % of the H₂/CO₂ selectivity can be recovered.”...

R1-7. As shown in the gas permeation results of CHF-850, the permeance of helium is lower than that of hydrogen, which is contradictory to molecular sieving mechanism.

A1-7: Although the gas transport through carbon membranes mainly depends on the penetrant dimension and molecule shape, the affinity and interactions with the carbon matrix can also affect the permeation behavior. The linear H₂ molecules may have a relatively low diffusion resistance compared to helium. That H₂ presents a higher permeance compared with helium has been reported in other molecular sieving membranes, such as, GO membranes (*Science* **342**, 95 (2013)). Mxene membranes (*Nature Communications* **9**, 155 (2018)), CMS membranes (*Advanced Materials* **29**, 1701631 (2017)). and *Carbon* **85**, 429-442 (2015), and other membrane types (*Nature Communications* **11**, 1633 (2020)).

R1-8. As shown in Raman spectra of obtained carbon hollow fiber membrane as function of pyrolysis temperature, it clearly observed the intensity of D1 peak arisen from CHF-850 maintained in comparison to other membranes at lower pyrolysis temperature, implying the packing defects still exist. However, the ratio of sp³/sp² decreased with rising the carbonization temperature as shown in XPS results, indicating the structure of carbon membrane become more ordered.

A1-8: The authors apologize for the unclear expression. The D1 peak in a Raman spectrum is the A_{1g}-symmetry vibration mode from the sp² carbon atoms in rings, disordered graphite, or defects in graphite (*Angewandte Chemie International Edition* **58**, 15089-15097 (2019)), (*Physical Chemistry Chemical Physics* **9**, 1276-1290 (2007)). Thus, the D1 peak still represents the sp² hybridized carbon but suggests some disorder in the sp² carbon, for example, the edges of the graphite crystallite (*Physical Chemistry Chemical Physics* **9**, 1276-1290 (2007)), (*Carbon* **150**, 142-152 (2019)). As also shown in the TEM images (**Fig. R4**), the carbon structure is graphitized by being carbonized at a higher temperature, resulting in a more ordered structure with a short range, which indicates the defects still exist in the carbon matrix. Thus, the Raman spectra, XPS results, and TEM images are consistent.

Action: More discussion about Raman spectra has been added in the revised manuscript:

P9: ... “The intensities of both the G and D1 peaks increase with carbonization temperature, which suggests a transformation towards a more graphitic carbon structure, consistent with the XPS results. But the strong D1 peak also indicates that some defects (such as the edges of the graphite^{21, 34}) still exist in the carbon matrix.”...

R1-9. The authors claimed the discrimination in size between ultra-micropore and micropore is 5 angstroms. However, according to the envisioned structure of carbon molecular sieving membrane as reported by William J. Koros et al., the size of micropores is ranging from 7-20 angstrom, while that of ultra-micropore is lower than 7 angstrom. I concern whether the difference in judging the kind of.

A1-9: Thanks for pointing this. The discrimination in the size between ultramicropores and micropores of 7 Å is widely used for CMS membranes in the literature. Considering the specific application of H₂/CO₂ separation in this work, the pore size of < 5 Å could have a significant contribution to selectivity based on the molecular sieving mechanism. As is also indicated in **Fig. 2d**, the PSD of < 5 Å is enhanced when carbonization temperature increases. However, a doublet peak located at the range of 4-7 Å may make the definition of the discrimination size at 5 Å unclear. Thus, to avoid any confusion, we have now removed the boundary distinction between ultramicropores and micropores in the PSDs of the original **Fig. 2d**.

Action: The boundary distinction between ultramicropores and micropores in the PSDs of the original **Fig. 2d** has been removed.

R1-10. The paper should be carefully checked and several missing unit of temperature (□) should be added.

(1) Grammatical errors at Page No. 6, Line No. 117: "...noting..." should be "noted"

(2) Missing units at the title of Fig. 1a

(3) Missing units at Page No. 14, Line No. 306

(4) Missing units at Line No. 104 in supplemental file

A1-9: Thanks for the corrections. We have carefully checked the manuscript and Supplementary Information and revised correspondingly. The authors would like to point out that the missing units in the manuscript and Supplementary Information might be caused by the PDF format conversion during submission.

Action: Grammatical and spelling errors have been revised in both manuscript and Supplementary Information.

Reviewer #2 (Remarks to the Author):

Reviewer R2-1. As the authors note, asymmetric cellulose-based carbon hollow fiber membranes have not been reported for gas separation. As the authors also note, solubility limitation require the use of rather exotic solvents such a as N-methylmorpholine-N-oxide (NMMO), ionic liquids and inorganic salts to achieve dopes here to allow flat sheet and hollow fibers here. This report extends their earlier work reported in I&ECR, 2019, 58, 13330-13339 paper (ref. 17) on screening of spinning parameters and applied primarily to the CO₂/CH₄ separation. While successful in terms of CO₂/CH₄ selectivity, the achievable permeability and permeance (permeability divided by selective layer thickness) were very low, and essentially impractical for that application.

A real advancement reported here that goes beyond the I&ECR paper is that the authors have now carefully optimized the quench bath temperature as a new variable to allow achieving asymmetry in the precursor. This additional variable allows maintaining some asymmetry in the CMS.

The CO₂ permeance, while higher than in the I&ECR paper is still quite low, but they now focus on H₂/CO₂ for hydrogen purification from the steam methane reforming process, rather than CO₂/CH₄ separation. In their paper they also note the ability to avoid treatments required for other precursors to avoid collapse for the morphology during pyrolysis.

I feel that the work is quite interesting but incomplete for such an application, since the steam-methane reforming process requires stability of the membrane to do the H₂/CO₂ separation in the presence of high temperatures and steam. If the authors will complement their current results with at least some results for ternary feeds comprising not only H₂ & CO₂, but also steam at temperatures up to 130 °C, I think the work would meet the standards of *Nature Communications*. On this basis, I note “major revisions” are required for the authors to accommodate this important issue. While it might be acceptable for the authors to only discuss and justify why they think the ternary results are not required, I think this is highly questionable and should be addressed with actual data.

Author A2-1 Answer: The authors would like to thank the reviewer for the positive comments and constructive suggestions.

We agree with the reviewer that conducting experiments under humidified conditions is relevant for investigating the potential application of the developed carbon membranes for H₂/CO₂ separation in a steam-methane reforming process. Thus, we have modified the high-pressure mixed gas permeation setup (as shown in Fig. R8), and the membrane module constructed by CHFM-700 was tested over 200 h under a 50 mol % H₂/50 mol % CO₂ mixed gas with 70% RH (relative humidity) and 100% RH at 10 bar and 90 °C, as shown in Fig. R9.

Fig. R8. Illustration of the high-pressure mixed gas permeation rig with humidity control.

Fig. R9. Mixed gas stability testing (50 mol % H₂/50 mol % CO₂) of CHFM-700 under dry and humidified conditions at 10 bar and 90 °C.

The membrane was initially examined using a dry mixed gas at a pressure of 10 bar for ca. 16 h. Then, the feed stream was switched to a humidified gas with 70% RH and tested over 20 hours. The presence of water vapor reduced the H₂ permeance from 95 GPU to 75 GPU and slightly reduced the H₂/CO₂ selectivity from 34 to 31. When the feed stream was changed back to a dry gas mixture, the H₂ permeance increased to 85 GPU. The membrane was then tested with 100% RH gas conditions over 120 h and showed stable separation performance. Such stable performance under humidified conditions can be attributed to the hydrophilic property of cellulose-derived CMS membranes. As shown in **Fig. R10**, the contact angles of water on the three type of CMS films are around 61-72 °. By increasing the carbonization temperature, the contact angle slightly increases, which is result of fewer hydrophilic -OH groups in the carbon matrix (see the answer of **A1-3** and **A1-4**).

Fig. R10. Contact angles of water on the CMS membranes prepared at different carbonization temperatures

The author would like to point out that the long-term stability testing has not been successfully conducted at a higher temperature (such as 130 °C) because of the following reasons:

1. Due to the pressure transducer, temperature sensor, and humidity indicator containing some sensitive electronic components, higher temperatures may cause damage of the circuits, since they are not rated for this temperature range.
2. In this work, the membrane module was sealed by Loctite EA 3430 epoxy adhesive. We found that the membrane modules sealed with this epoxy potting material cannot withstand for a long time the conditions under water vapor contained gas stream at 130 °C and 10 bar.

In our future work, we will try to identify a more suitable epoxy potting material for the specific testing condition.

Action: A membrane module of CHF700 tested under humidified conditions has been discussed in the revised manuscript. The **Fig. R9** was added in **Fig. 3** in the revised manuscript. More discussion about the humidified conditions and hydrophilicity were included in the revised manuscript:

P13: ...“The CHF700 was further assessed by dynamic durability testing over 200 h under a 50 mol % H₂/50 mol % CO₂ mixed gas with 70% RH (relative humidity) and 100% RH at 10 bar and 90 °C. As shown in **Fig. 3e**, the membrane was initially tested using a dry mixed gas at a pressure of 10 bar for ca. 16 hours. Then, the feed stream was switched to a humidified gas with 70% RH and tested over 20 h. The presence of water vapor reduced the H₂ permeance from 95 GPU to 75 GPU and slightly reduced the H₂/CO₂ selectivity from 34 to 31. When the feed stream was changed back to a dry gas mixture, the H₂ permeance increased to 85 GPU. The membrane was then tested at 100% RH gas conditions over 120 h and showed stable separation performance. Such stable performance under humidified conditions can be attributed to the hydrophilic property of cellulose-derived CMS membranes. As shown in **Supplementary Fig. 19**, the contact angles of water on the three types of CMS films are around 61-72°. By increasing carbonization temperatures, the contact angle slightly increases, which is the result of fewer hydrophilic -OH groups in the carbon matrix” ...

P15: ...“The hydrophilic carbon membranes of CHF700 show good stability for over 120 h under 100% RH humidified mixed gas at 10 bar and 90 °C, which indicates the great potential for H₂ purification in steam reforming processes.”

Reviewer #3 (Remarks to the Author):

The paper does not report a striking novelty – but the healthy development of a practice-relevant carbon membrane with a chance of commercialization. By a clever selection of the starting materials and sophisticated carbonization conditions, self-supporting carbon hollow fiber membranes with asymmetric cross sections have been developed. The characterization by XPS (sp^2 - sp^3) and Raman is correct, but not new. After balancing the different opinions, I prefer a publication of the manuscript with a 60-40 opinion. Since the manuscript is well prepared and the techniques are well described, there is not much space at the bottom and here are only a few remarks for a revision.

Author Answer: The authors are grateful for the reviewer's positive comments.

Reviewer R3-1. Fibers can be bent with a radius of 1.5 cm. Permeation after bending? there further mechanical properties (Young module) of the self-supporting hollow fibers?

Author Answer 3-1: In order to evaluate the separation performance of a selected carbon hollow fiber membrane (CHFM-850) module, the membrane module for single gas permeation testing (see **Fig. R11a**) was bent with a diameter of 3.7 cm (**Fig. R11b**), and its separation performances were found the same.

Fig. R11. Photographs of a membrane module being bent with a diameter of 3.7 cm, and used for single gas permeation tests.

The Young's modulus was measured by nanoindentation tests using a Berkovich indenter. The CHFM samples were loaded to the maximum load ($P_{max} = 1000 \mu\text{N}$) in 5 s and then held for 2 s, followed by unloading in 5 s. The resulting load–displacement curves are shown in **Fig. R12**. The hardness, reduced elastic modulus, and Young's modulus are listed in **Table R1**.

Fig. R12. Load–displacement curves of CHFMs from nanoindentation tests.

Table R1 Nanoindentation hardness, reduced modulus, and Young’s modulus of CHFMs

Samples	Hardness (GPa)	Reduced modulus (GPa)	Young’s modulus (GPa)
CHF-550	0.31±0.06	2.16±0.26	2.07±0.25
CHF-700	0.84±0.25	5.93±0.06	5.69±0.07
CHF-850	1.30±0.10	7.85±0.50	7.53±0.48

Action: **Fig. R11, R12,** and **Table R1** were added in the Supplementary Information. Nanoindentation tests for CHFMs have been included in the revised manuscript and Supplementary Information:

In the revised manuscript:

P7: ...“The Young’s modulus of prepared CHFMs measured by nanoindentation tests is summarized in **Supplementary Table 2** and load–displacement curves are shown in **Supplementary Fig. 7**. Like other carbonized carbon materials^{29,30}, the CHF-550 prepared at the lowest carbonization temperature exhibits the deepest displacement by indentation, leading to the lowest hardness (0.31 GPa) and Young’s modulus (2.07 GPa). As the carbonization temperature increases to 850°C, the hardness and Young’s modulus increase to 1.30 GPa and 7.85 GPa, respectively. The enhanced hardness and modulus can be attributed to the change of internal structure by raising carbonization temperature, such as the increased sp²-hybridized bonds in carbon²⁹. ”...

P17-18:“Nanoindentation tests were conducted by TriboIndenter 950 by using a Berkovich indenter (details are given in Supplementary Note 4)”...

In the revised Supplementary Information:

P4: ...“**Supplementary Note 4. Nanoindentation test**

The hardness, the reduced elastic modulus, and Young’s modulus of CHFMs were measured by nanoindentation tests using a Berkovich indenter. The CHF-550 samples were loaded to the maximum load ($P_{max} = 1$ mN) in 5 s and then held for 2 s, followed by unloading in 5 s. The measured hardness and reduced elastic modulus are summarized in **Supplementary Table 2**. The Young’s modulus (E_r , GPa) were estimated using the Oliver–Pharr method¹ as follows:

$$\frac{1}{E_r} = \frac{1-\nu^2}{E} + \frac{1-\nu_{tip}^2}{E_{tip}} \quad (S1)$$

Where ν and E are Poisson’s ratio and Young’s modulus of the CHF-550 samples, respectively. ν_{tip} and E_{tip} are Poisson’s ratio and Young’s modulus of the indenter, respectively. $\nu_{tip} = 0.07$ and $E_{tip} = 1140$ GPa. Poisson’s ratio of CHFMs is assumed to be the same and equal to 0.2². Since $E_{tip} \gg E_r$, the

second term of the equation S1 is negligible. Hence, the Young's modulus of the samples is approximated to $E = 0.96 E_r$. "...

R3-2. It is common that membrane people use Robeson's upper bound from 2008 as a benchmark like a holy grail. But this was 12 years ago. Are there new developments in the world of polymer membranes in the last 12 years?

A3-2: Yes, there have been significant developments of polymer membranes that show good separation performances and surpass the Robeson Upper bound 2008, for example, reported by Zhu. et al. (*Energy & Environmental Science* **11**, 94-100 (2018)), by Shan et al. (*Science Advances* **4**, eaau1698 (2018)), , by Rose et al. (*Nature Materials* **16**, 932 (2017)), by Song et al. (*Nature Communications* **5**, 4813 (2014), *Nature Communications* **4**, 1918 (2013)), and by Liu et al. (*Nature Communications* **11**, 1633 (2020)).

The upper bounds for some gas pairs have also been updated and redefined. The CO₂/N₂ and CO₂/CH₄ upper bounds have been redefined by Comesaña-Gándara et al. (*Energy & Environmental Science* **12**, 2733-2740 (2019)). New 2015 upper bounds of O₂/N₂, H₂/N₂, and H₂/CH₄ are reported by Swaidan et al. (*ACS Macro Letters* **4**, 947-951 (2015)). In our original Supplementary Information, both the 2008 and 2015 upper bounds of H₂/N₂ and H₂/CH₄ were used.

However, H₂/CO₂ separation is quite challenging. Many inorganic membranes cannot achieve the precise cut-off such that the pores can discriminate the small differences in the CO₂ and H₂ molecular size. For polymeric membranes, even though H₂ presents a higher diffusion coefficient, it has a smaller solubility coefficient compared to CO₂, which leads to an overall low H₂/CO₂ selectivity (usually <10). Increasing operating temperature can potentially reduce CO₂ solubility coefficient and enhance H₂/CO₂ selectivity, but most of polymeric membranes cannot withstand high temperature operation (e.g., steam methane reforming integrated with water gas shift process at a temperature of >150 °C). Therefore, high performance membranes for H₂/CO₂ separation at high pressure and temperature operation are less developed, and the Robeson 2008 upper bound of H₂/CO₂ is still used in the current work and elsewhere as a benchmark.

R3-3: Authors say that the carbon membranes are cheap using cellulose. But also an IL is used. The price of the IL depends on its purity and is between 300 and 1000 €/kg. Is this a problem?

A 3-3: Yes. The price of ILs is relatively expensive compared to conventional solvents. Thus, for large-scale membrane production, the recovery of ILs is required. Different methods have been reported, such as freeze crystallization (Liu et al, *Green Chemistry* **20**, 493-501 (2018)) and membrane separation (Lynam et al, *Chemical Engineering Journal* **288**, 557-561 (2016)). Those methods can be considered for the recovery of ILs. The authors would like to point out that the carbon membranes developed in this work are based on a renewable and low cost polymer of cellulose, and the employed preparation process is facile, and simple without the requirement of additional pre-treatment to prevent pore collapse, which in our opinion can offset the high cost of ILs, and provide a competitive membrane production cost compared to other carbon membranes reported in the literature.

Action: the following statement has been included in the revised manuscript:

P15: ...“The carbon membranes developed in this work are based on a renewable and low cost polymer of cellulose, and the employed preparation process is facile, and without the requirement of additional pre-treatment to prevent pore collapse, which provides a competitive membrane production cost compared to other carbon membranes, particularly when the ILs are recovered.”

R3-4. Some graphics need improvements:

Fig. 3a: H₂/CO₂ Selectivity contra Fig. 4: H₂/CO₂ selectivity

Fig. 3a: H₂ Permeance contra Fig. 4: H₂ permeability

A3-4: Thanks for the suggested corrections. We have carefully checked graphics and revised correspondingly. For example, Fig. 3a H₂/CO₂ Selectivity was revised to H₂/CO₂ selectivity, Fig. 3a H₂ Permeance was revised to H₂ permeance. Fig. S14. Ln Permeability was revised to ln permeability.

R3-5. Apply Dunitz’ rule: The first sentence of every manuscript can be deleted, e.g. “Fossil fuels are limited / Cancer is a plague / Global warming is a threat to humanity”

A 3-5: Thanks for the good suggestion. We have removed the first sentence.

R3-6. Can the authors give some application-relevant remarks: How can these fibers be fixed in a module? Regeneration? Scale up of the fiber fabrication?

A 3-6: As illustrated in **Fig. R13**, the CHFMs are normally constructed in a steel tubing. In this work, several CHFMs are fixed in a Swagelok 3/8-inch tubing for mixed gas testing (**Fig. R13a**). Both sides are sealed by epoxy adhesive (Loctite EA 3430). If membrane modules are only used for single gas permeation testing and no sweep gas is needed, dead-ended hollow fibers were made by blocking one end with epoxy adhesive, as shown in **Fig. R13b (right side)** and **Fig R11 (upside)**.

Thermal and chemical treatments can be used for the regeneration of CMS membranes. In this work, we have tried thermal regeneration as we commented in **A1-6**. Membranes exposed to propylene regeneration has been reported by Kumar and Koros (*Industrial & Engineering Chemistry Research* **58**, 6740-6746(2019)).

For membrane scale-up, a pilot CHFMs membrane module (the membranes made from cellulose acetate) with 3000 fibers and 0.8 m length has been reported (*Separation and Purification Technology* **190**, 177-189 (2018)). However, scaling-up of CHFMs prepared from cellulose and ionic liquids may still face the following challenges, and the potential solutions are listed correspondingly.

1. The relatively high cost of EmimAc as the reviewer mentioned. Reducing the ratio of EmimAc/DMSO in dope solution and recovering EmimAc can be applied.
2. It is crucial to drain the tars and remove vapors during the carbonization if large amounts of fibers are carbonized in a furnace. By setting a small angle (e.g., 6°) between the quartz support and furnace can be used for draining tars.
3. Membrane module design and construction are also important, such as CHFMs mounting, potting and sealing. Due to the H₂/CO₂ separation are often used under high -temperature and -pressure conditions, a better potting material (compared to epoxy resin used in this work) that can endure the humidified gas at high pressure and temperature should be identified.

Fig. R13. Schematic of membrane module for carbon hollow fiber membranes, a) for mixed gas testing, and b) for single gas testing.

Action: This has now been included in the manuscript and Supplementary Information.

In the revised manuscript:

P17: ...“The schematics of the construction of carbon membrane modules is illustrated in **Supplementary Fig. 22**. In brief, several CHFMs are mounted in a Swagelok 3/8-inch tubing, and both sides are sealed by epoxy adhesive (Loctite EA 3430). If membrane modules are only used for single gas permeation testing and no sweep gas is needed, dead-ended hollow fibers were made by blocking one end of fibers with epoxy adhesive, as shown in **Supplementary Fig. 22b**. The **supplementary Fig. 23** gives the flexibility of the membrane module for single gas testing, which is bent with a diameter of 3.7 cm. It was found that the separation performances were maintained after bending.”...

P17: ...“Possible issues for membrane fabrication scale-up and potential solutions are discussed in **Supplementary Note 7**.”...

In the revised Supplementary Information:

P7: Supplementary Note 7. Possible issues for membrane fabrication scale-up

Scaling-up of CHFMs prepared from cellulose and ionic liquids may still face the following challenges, and the potential solutions are listed correspondingly.

1. Reducing the ratio of EmimAc/DMSO in dope solution and recovering EmimAc can be applied to bring down the relatively high cost of ionic liquids.
2. It is crucial to drain the tars and remove vapors during the carbonization if large amounts of fibers are carbonized in a furnace. By setting a small angle (e.g., 6°) between the quartz support and furnace can be used for draining tars.
3. Membrane module design and construction are also important, such as CHFMs mounting, potting and sealing. Due to the H₂/CO₂ separation are often used under high -temperature and -pressure conditions, a better potting material (compared to epoxy resin used in this work) that can endure the humidified gas at high pressure and temperature should be identified.

REVIEWER COMMENTS

Reviewer #1 (Remarks to the Author):

#NCOMMS-20-18017A-Carbon Molecular Sieve Hollow Fiber Membranes with Precise-cutoff Ultramicropores for Superior Hydrogen Separation

The authors had addressed most of questions aroused from reviewers. However, it still has several questions in the revised manuscript to be resolved before acceptance for publication.

1. In view of the author response (AR1-1), the difference of cellulose-derived carbon hollow fiber membrane (CHFM) in this work, in compared with other literatures has been stated. However, the reason for overwhelming H₂/CO₂ performance of cellulose-derived CHFM compared with same precursor is still not mentioned.
2. Regarding to above comment, the author explained the higher selectivity of cellulose-derived CHFM for H₂/CO₂ is because of the absence of the heteroatoms, such as N and F. The cited paper merely mentioned the presence of heteroatoms could enhance the sorption property of carbon dioxide. Therefore, how to conclude the lack of heteroatom in the adopted precursor has limited CO₂ affinity. Could the author explain more? In addition, as mentioned by author, a low heteroatom-content CMS membrane possesses inferior CO₂-philic property. The pore size distribution (PSD) is detected by CO₂ physisorption measurement. The correctness of obtained PSD data from CMS material with lower CO₂-philic property whether is affected or not.
3. The d-spacing value of CHFM-850 membrane is 3.50 angstrom, while the kinetic diameter of CO₂ is 3.2 angstrom. Why the H₂/CO₂ selectivity of CHFM-850 is the best?
4. As shown in Supplementary Fig. 14, when the pyrolysis temperature increased to 850 °C, the peaks of H₂O and CO₂ were detected and thus the author noted the O and H atoms had been removed further. However, the XPS spectra (Supplementary Fig. 11) shows the crest of O atom was detected and the peak area did not decreased with increasing the pyrolysis temperature from 550 °C to 850 °C, while XPS is semi-quantitative analysis. Moreover, "spectrum" should be used in singular noun. As a result, the title of Supplementary Fig. 11 should be revised.

Reviewer #2 (Remarks to the Author):

I was positive about the previous paper, EXCEPT for one major deficiency—the lack of high temperature ternary H₂/CO₂/H₂O (steam) feed separation results. In the current paper, the authors have made a good attempt to satisfy this concern.

I am impressed by the significant effort the authors made to address this important requirement. Even though the authors did not test at 130 C °as I had requested, they did test at 90 C with both dry and wet feeds with humidity and report the results in Fig R9 and incorporated in Fig. 3.

While the H₂ permeances are low, since the membranes are in hollow fiber forms, they can still pack a lot of membrane area in a compact module, so this deficiency is not so bad.

On the basis of the above assessment, I think it is reasonable to accept the paper without further revision.

Reviewer #3 (Remarks to the Author):

The authors did a very careful revision. I can accept their arguments and recommend ACCEPT.

There are 2 minor editorial problems, but I am not sure to send the manuscript back because of these tiny problems.

Figure 3d) The right ordinate "H₂/CO₂ selectivity" and the curve are both in blue. The left ordinate "Permeance" is in black, but the curve in orange.

Figure 3a: "H₂ Permeance", but Fig. 4 "H₂ permeability"

RE: Point-by-point response for manuscript NCOMMS-20-18017A

Title: Carbon Molecular Sieve Hollow Fiber Membranes with Precise-cutoff Ultramicropores for Superior Hydrogen Separation

Authors: Linfeng Lei, Fengjiao Pan, Arne Lindbråthen, Xiangping Zhang, Magne Hillestad, Yi Nie, Lu Bai, Xuezhong He, Michael D. Guiver

Response to Reviewers' comments:

The authors again thank the reviewers for their valuable comments and efforts to improve our manuscript. We highly appreciate their positive feedback on our revised manuscript. We have carefully considered all the reviewers' comments and have revised the manuscript to address their concerns. To aid in the reviewing process, we have replied to all the comments on a point-by-point basis and highlighted the revised sections of the main manuscript and Supplementary Information with red font color. We hope the manuscript is now in an acceptable form for publication in *Nature Communications*.

On behalf of the authors, and with kind regards,

Dr. Michael D. Guiver (designated corresponding author for the submission)

Dr. Xuezhong He (co-corresponding author)

Reviewer #1 (Remarks to the Author):

#NCOMMS-20-18017A-Carbon Molecular Sieve Hollow Fiber Membranes with Precise-cutoff Ultramicropores for Superior Hydrogen Separation

Reviewer: The authors had addressed most of questions aroused from reviewers. However, it still has several questions in the revised manuscript to be resolved before acceptance for publication.

Author Answer: Thank you very much for your positive feedback on our revised manuscript. Here we reply to your comments below:

R1-1: In view of the author response (AR1-1), the difference of cellulose-derived carbon hollow fiber membrane (CHFM) in this work, in compared with other literatures has been stated. However, the reason for overwhelming H₂/CO₂ performance of cellulose-derived CHFM compared with same precursor is still not mentioned.

A1-1: The previously reported CHFMs made from cellulose-based precursors (either the regenerated cellulose from cellulose acetate¹ or microcrystalline cellulose^{2,3}) have been mainly reported for CO₂/N₂ or CO₂/CH₄ separations, not for H₂/CO₂ separation. Thus, the target in those previous studies was to prepare CO₂-selective and CO₂-permeable membranes, which requires slightly larger pores that allow the smaller CO₂ molecules to permeate through the membranes but exclude the larger nitrogen and methane molecules. In that case, the pore size distribution (PSD) of the prepared membranes are normally located in the range 4.5-7Å (ultramicropores) and 7-10 Å (micropores). However, our work focuses on H₂/CO₂ separation (i.e., exclusion of CO₂ while allowing the smaller H₂ molecule to pass through). To achieve this gas pair separation, the required pore sizes of membranes must be smaller than the previous studies, which increases the CO₂ transport resistance, thereby improving H₂/CO₂ diffusion selectivity. Thus, compared with the previously reported cellulose-derived CMS membranes, the difference of the asymmetric CHFMs obtained in this work is that the pore size is tuned to greatly limit the permeation of CO₂ molecules, which are larger than H₂ molecules.

The authors would like to point out that although the increase of carbonization temperature has been reported to narrow the pore size thereby enhancing gas pair selectivity in the literature, gas permeance was usually found to decrease significantly, showing a strong trade-off between gas permeance and selectivity. For example, a higher carbonization temperature (increased from 750 to 900 °C) gave a H₂/CO₂ selectivity, which increased from ~3 to 11, as reported by the Koros group⁴. However, there was a large concomitant decrease in H₂ permeability from 1600 to 240 barrer, which corresponds to a reduction in H₂ permeance from 65 to 10 GPU (calculated from a selective layer of ca. 25 μm). Therefore, it is expected that a large decrease in gas permeance will occur at higher carbonization temperatures, if we use the same symmetric cellulose hollow fiber precursors as reported in previous studies.

In our work, we controlled the precursor structure to successfully obtain an asymmetric structure with a selective layer thickness of ca. 3 μm to offset the permeance drop when membranes are prepared at a higher carbonization temperature. As we have clearly stated in the manuscript, one of the novelties of our work is that we obtained asymmetric CHFMs from asymmetric cellulose hollow fiber precursor, which is reported for the first time. This procedure does not require any additional crosslinking treatment, which has often been used in the preparation of asymmetric CMS membranes to avoid pore collapse.

On the other hand, the hollow fiber spinning conditions affect the properties of the membrane precursors. The selective layer of cellulose hollow fibers obtained at a higher coagulation temperature (60 °C) in this work is denser compared with the previously reported symmetric cellulose precursors (using the same cellulose material) coagulated at room temperature. For example, the ultramicropore size of the CHFMs-550, even though it is carbonized at 550 °C, is smaller compared with the previously reported carbon

membranes carbonized at 600 °C having pore size $>4.5 \text{ \AA}^2$. Thus, the property changes of the spun asymmetric cellulose precursors by using a higher coagulation temperature and a nonsolvent pretreatment (to avoid pore collapse), contribute to reducing and controlling the pore size compared with the previous symmetric cellulose-based membranes.

The obtained asymmetric carbon membranes with a much thinner selective layer reduce the gas transport resistance compared with the previously reported symmetric carbon membranes with a typical wall-thickness of $>20 \text{ \mu m}$. Therefore, the obtained asymmetric precursors in this work have a crucial role in providing carbon membranes with overwhelming H_2/CO_2 separation performance (especially H_2 permeance, while the selectivity was further tuned by the carbonization temperature) that can exceed the Robeson upper bound, beyond the carbonization conditions.

Action: The following paragraph has now been included in the revised manuscript to emphasize the difference between the prepared asymmetric CHFMs and the reported symmetric CHFMs:

...“The cellulose-based CMS membranes reported high gas selectivities for O_2/N_2 ²⁰ and CO_2/CH_4 ¹⁹. The asymmetric cellulose-based CHFMs developed in our work present a new approach for light gas separations by creating more ultramicropores, by a combination of the coagulation temperature and nonsolvent exchange assisted morphological control during fiber spinning, as well as by tuning the carbonization temperature.”...

R1-2: Regarding to above comment, the author explained the higher selectivity of cellulose-derived CHFMs for H_2/CO_2 is because of the absence of the heteroatoms, such as N and F. The cited paper merely mentioned the presence of heteroatoms could enhance the sorption property of carbon dioxide. Therefore, how to conclude the lack of heteroatom in the adopted precursor has limited CO_2 affinity. Could the author explain more? In addition, as mentioned by author, a low heteroatom-content CMS membrane possesses inferior CO_2 -philic property. The pore size distribution (PSD) is detected by CO_2 physisorption measurement. The correctness of obtained PSD data from CMS material with lower CO_2 -philic property whether is affected or not.

A1-2: This is a good comment. We apologize for this unclear statement. In this work, the CO_2 sorption capacity of carbon membranes was reduced by the increase of the carbonization temperature, as illustrated in **Supplementary Fig. 10**. The cited papers we provided discussed CO_2 sorption changes by the doped heteroatoms. Thus, we have now revised the words “ CO_2 affinity” to “ CO_2 sorption capability”

to make the meaning clearer. Many papers have demonstrated that an increase of nitrogen content in the carbon materials could enhance the CO₂ sorption capability^{5, 6, 7}, and thereby improve CO₂ capture capacities. In those studies, the higher CO₂ sorption capacity is preferable because they are targeting CO₂-selective membranes (or adsorbents). However, our current work is not targeting CO₂-selective membranes. Instead we are targeting H₂/CO₂ separation, and high CO₂ uptake should be avoided. For example, previous studies reported CO₂ uptake of ca. 6 mmol g⁻¹⁶ and 5 mmol g⁻¹⁸ (at 273K, 1 bar), which is higher than the CO₂ uptake reported in our work (i.e., 2.6 mmol g⁻¹ for CHFM-850, **Supplementary Fig. 10a**, converting to mmol g⁻¹ from cm³ g⁻¹). Although most of the reported work employed nitrogen-contained sources to synthesize the porous carbon materials, hardly any work has documented the sorption capability changes by removing heteroatoms. The authors suspect that the heteroatoms in carbon matrix can cause defects in the π -cloud generated by the sp² hybridized carbons. Hence, it disrupts the packing of the graphene sheets and thus changes the pore size of carbon matrix. Based on the previously reported work that the increase of heteroatoms (apart from O) content can enhance the CO₂ sorption capability, we believe it is reasonable to infer that carbon membranes derived from precursors with no or low content of heteroatoms should result in exhibit reduced CO₂ sorption.

We thank the reviewer for the valuable comment again. In future work, investigation on heteroatom-doped cellulose-based carbon membranes can be pursued to understand the relationship between CO₂ sorption capability and the elemental contents of porous carbon materials.

To avoid any unclear statement in the previous manuscript, we have now revised the sentences as follows,

“It has been reported that increasing the doping content of heteroatoms, such as N, into carbon materials could enhance the CO₂ sorption capabilities, thereby providing highly CO₂-selective materials^{17,18}. To et al.¹⁷ demonstrated Henry’s Law selectivity of CO₂/N₂ for N-doped porous carbons increased from 9 to 124 when the N content was increased from 3.2 to 5.8 wt.%. Similarly, Yang et al.¹⁸ reported that the diffusion of CO₂ molecules through the membrane was improved by N- and F-containing nanodomains. As a corollary, it is inferred that CMS materials with a low heteroatom-content should reduce CO₂ sorption, and may provide an approach to achieve highly H₂/CO₂ selective carbon membranes.”

The calculation of PSD of carbon membranes using CO₂ as adsorbent based on a nonlocal density functional theory (NLDFT) model (CO₂ - carbon equilibrium transition kernel at 273K on a slit-pore model⁹) have been widely reported. In the slit-pore model, the individual pore is represented as two parallel smooth graphite, and the carbon-CO₂ interaction potential is described by the Steele (10-4-3) Potential^{9,10,11},

$$\phi_{sf}(z) = 2\pi\varepsilon_{sf}\rho_s\sigma_{sf}^2\Delta \left[\frac{2}{5} \left(\frac{\sigma_{sf}}{2} \right)^{10} - \left(\frac{\sigma_{sf}}{2} \right)^4 - \frac{\sigma_{sf}^4}{3\Delta(z+0.61\Delta)^3} \right] \quad \text{R1}$$

Where z is the distance from the graphite surface, Δ is the distance between layers in graphite (0.335 nm), ρ_s is the number density of graphite. ε_{sf} and σ_{sf} are the fluid-solid Lennard-Jones parameters.

Thus, in the NLDFT model, the carbon-CO₂ interaction is kept the same for different samples. Although the model can be improved by introducing structural and/or energetical heterogeneity to the surface of pore walls^{12, 13}, the NLDFT model is considered as a common method to estimate the PSD of carbon membranes. Thus, we believe that the change of CO₂ sorption capacity would not affect the correctness of the obtained PSD. But the CO₂ sorption capability can be affected by surface area and micropore volume, and thus obtain the different PSDs for different CHFMs.

R1-3: The d-spacing value of CHFM-850 membrane is 3.50 angstrom, while the kinetic diameter of CO₂ is 3.2 angstrom. Why the H₂/CO₂ selectivity of CHFM-850 is the best?

A1-3: This is a good comment. Although the d-spacing of 3.5Å for the CHFM-850 obtained from XRD analysis is larger than the kinetic diameter of CO₂, the obtained superior H₂/CO₂ selectivity of CHFM-850 of 83.9 is attributed to its unique microstructure,

1). Since our carbon membrane has graphite-like structure with sp³-carbon defects and a disrupted packing of the graphene sheets, as shown in the HRTEM (**Fig. 2g**) and indicated by Raman spectra, the calculated d-spacing is ca. 3.50 Å based on Bragg's law from XRD, which is slightly larger than the ideal d-spacing of graphite (3.35 Å). This is because of the existence of some defects (such as, the carboxylic acid, carbonyl, and epoxy groups). However, these polar groups located between the ordered layers can retard CO₂ transport. This phenomenon was reported by Kim et al.¹⁴, who observed that the carboxylic acid groups in GO membranes can trap CO₂ molecules and thus strongly retard CO₂ transport. This is because that the polarity of the individual C=O bonds in the CO₂ molecule allows for interaction with polar groups in GO. Thus, CO₂ can act as a Lewis acid or a Lewis base and can participate in hydrogen bonding¹⁴.

2). In contrast with other two-dimensional nanosheets, our carbon membranes have an effective selective layer consisting of ultramicropores and micropores for gas permeation (e.g., 3 μm thickness, but not a few-layered nanosheet). Thus, the overall transport resistance is not only in the interlayers (ultramicropores that determine diffusivity), but also the imperfectly stacked “carbon plates” (micropores that determine solubility coefficient).

3) One should expect that some CO₂ molecules can still pass through the CHF_M-850 membrane based on the measured CO₂ permeance. If it was possible to fabricate a carbon membrane where all the pore sizes were smaller than 3.3 Å, the theoretical H₂/CO₂ selectivity should be extremely high, because the CO₂ molecules would be fully rejected. More frequent collisions between the CO₂ molecules and the pore wall are expected when CO₂ molecules transport through the confined pore channels with the size of smaller than 3.5 Å, which increases the resistance and time of passage, which leads to a lower CO₂ diffusivity for the CHF_M-850. Thus, the smaller d-spacing of CHF_M-850 compared with the other two carbon membranes presents a longer retention time for CO₂ transportation, while H₂ permeance has less affected due to its substantially smaller kinetic diameter of 2.9 Å. Therefore, the H₂/CO₂ diffusion selectivity of CHF_M-850 has the best value. On the other hand, due to the reduced pore size and PSD (**Fig. 2d**), the CO₂ adsorption capacity is also reduced for the CHF_M-850 as shown in the Supplementary Fig. 10, while the H₂ adsorption is less affected. Thus, a higher solubility selectivity of H₂/CO₂ is also expected for CHF_M-850.

Based on the above three points, **the CHF_M-850 present the best H₂/CO₂ selectivity.**

Action: More discussion is included in the revised manuscript:

...“Although the d-spacing of CHF_M-850 (**Fig. 2c**) is 3.5 Å, which is larger than the CO₂ kinetic diameter of 3.2 Å, the achieved highest H₂/CO₂ selectivity of 83.9 is the result of its unique microstructure. As illustrated in **Supplementary Fig. 13**, the existence of oxygen containing functional groups (such as carboxylic acid, carbonyl, and epoxy groups) between the ordered layers can strongly retard CO₂ transport, as reported by Kim et al.⁵ Besides, the...⁵”...

R1-4: As shown in Supplementary Fig. 14, when the pyrolysis temperature increased to 850 °C, the peaks of H₂O and CO₂ were detected and thus the author noted the O and H atoms had been removed further. However, the XPS spectra (Supplementary Fig. 11) shows the crest of O atom was detected and the peak area did not decrease with increasing the pyrolysis temperature from 550 °C to 850 °C, while XPS is semi-quantitative analysis. Moreover, “spectrum” should be used in singular noun. As a result, the title of Supplementary Fig. 11 should be revised.

A1-4: Thank you for pointing this. The title of **Supplementary Fig. 11** has been revised correspondingly. According to thermogravimetric analysis-mass spectrometry (TGA-MS) results, as shown in **Supplementary Fig. 14**, the H₂O and CO₂ peaks are detected, which means that more O atoms were removed at a higher carbonization temperature. Perhaps it is difficult for the reviewer to see the trend of reduced O content from the full XPS spectra because of its small peak. Thus, the O1s XPS spectrum for

the three different CHFMs is now included in the revised Supplementary Information (**Supplementary Fig. 14b**, also see **Fig. R1b**). Besides, in **Supplementary Table 3** (Elemental composition of the CHFMs from XPS analysis), which was included in the previous Supplementary Information, it was already shown that the oxygen content was reduced from 9.26 % to 7.04 % when the carbonization temperature was increased from 550 to 850 °C. This is now listed in **Table R1**.

Fig. R1. a) XPS survey spectra, and b) O 1s spectra for three different CHFMs

Table R1 Elemental composition of the CHFMs from XPS analysis.

	C (Atomic %)	O (Atomic %)	N (Atomic %)
CHFMs-550	90.08	9.26	0.67
CHFMs-700	91.25	8.10	0.65
CHFMs-850	92.41	7.04	0.55

Action: The title of **Supplementary Fig. 11** is changed to “XPS survey spectra for three different CHFMs”. The following sentence is included in the revised manuscript,

...“ both the observed peaks of H₂O (at ~ 800 °C) and CO₂ (at ~ 850 °C) and the reduced content of oxygen detected by XPS (**Supplementary Fig. 11b** and **Supplementary Table 3**) imply that the O and H atoms were further removed, and short-range ordered graphitic sheets were formed, as depicted in **Fig. 2g**.”...

References

1. He X, Lie JA, Sheridan E, Hagg M-B. Preparation and Characterization of Hollow Fiber Carbon Membranes from Cellulose Acetate Precursors. *Ind Eng Chem Res* **50**, 2080-2087 (2011).

2. Lei L, Lindbråthen A, Hillestad M, Sandru M, Favvas EP, He X. Screening Cellulose Spinning Parameters for Fabrication of Novel Carbon Hollow Fiber Membranes for Gas Separation. *Ind Eng Chem Res* **58**, 13330-13339 (2019).
3. Lei L, *et al.* Preparation of carbon molecular sieve membranes with remarkable CO₂/CH₄ selectivity for high-pressure natural gas sweetening. *J Membr Sci* **614**, 118529 (2020).
4. Zhang C, Koros WJ. Ultraselective Carbon Molecular Sieve Membranes with Tailored Synergistic Sorption Selective Properties. *Adv Mater* **29**, 1701631 (2017).
5. Hao G-P, Li W-C, Qian D, Lu A-H. Rapid Synthesis of Nitrogen-Doped Porous Carbon Monolith for CO₂ Capture. *Adv Mater* **22**, 853-857 (2010).
6. To JWF, *et al.* Hierarchical N-Doped Carbon as CO₂ Adsorbent with High CO₂ Selectivity from Rationally Designed Polypyrrole Precursor. *J. Am. Chem. Soc.* **138**, 1001-1009 (2016).
7. Kim YK, Kim GM, Lee JW. Highly porous N-doped carbons impregnated with sodium for efficient CO₂ capture. *J. Mater. Chem. A* **3**, 10919-10927 (2015).
8. Yang Z, *et al.* Surpassing Robeson Upper Limit for CO₂/N₂ Separation with Fluorinated Carbon Molecular Sieve Membranes. *Chem* **6**, 631-645 (2020).
9. Lastoskie C, Gubbins KE, Quirke N. Pore size distribution analysis of microporous carbons: a density functional theory approach. *J. Phys. Chem.* **97**, 4786-4796 (1993).
10. Ravikovitch PI, Vishnyakov A, Russo R, Neimark AV. Unified Approach to Pore Size Characterization of Microporous Carbonaceous Materials from N₂, Ar, and CO₂ Adsorption Isotherms. *Langmuir* **16**, 2311-2320 (2000).
11. Stecki J, Steele (10-4-3) Potential due to a Solid Wall. *Langmuir* **13**, 597-598 (1997).
12. Jagiello J, Olivier JP. 2D-NLDFT adsorption models for carbon slit-shaped pores with surface energetical heterogeneity and geometrical corrugation. *Carbon* **55**, 70-80 (2013).
13. de Oliveira JCA, *et al.* On the influence of heterogeneity of graphene sheets in the determination of the pore size distribution of activated carbons. *Adsorption* **17**, 845-851 (2011).
14. Kim HW, *et al.* Selective Gas Transport Through Few-Layered Graphene and Graphene Oxide Membranes. *Science* **342**, 91 (2013).

Reviewer #2 (Remarks to the Author):

Reviewer R2-1: I was positive about the previous paper, EXCEPT for one major deficiency—the lack of high temperature ternary H₂/CO₂/H₂O (steam) feed separation results. In the current paper, the authors have made a good attempt to satisfy this concern.

I am impressed by the significant effort the authors made to address this important requirement. Even though the authors did not test at 130 C as I had requested, they did test at 90 C with both dry and wet feeds with humidity and report the results in Fig R9 and incorporated in Fig. 3.

While the H₂ permeances are low, since the membranes are in hollow fiber forms, they can still pack a lot of membrane area in a compact module, so this deficiency is not so bad.

On the basis of the above assessment, I think it is reasonable to accept the paper without further revision.

Author Answer A2-1: We thank the reviewer for approving that we have correctly amended the manuscript. We also thank the reviewer for recommendation of our manuscript for publication.

Reviewer #3 (Remarks to the Author):

Reviewer R3-1: The authors did a very careful revision. I can accept their arguments and recommend ACCEPT.

Author Answer A3-1: Thank you very much for your recommendation of our work and your favorable comments on the manuscript.

There are 2 minor editorial problems, but I am not sure to send the manuscript back because of these tiny problems.

R3-2: Figure 3d) The right ordinate "H₂/CO₂ selectivity" and the curve are both in blue. The left ordinate "Permeance" is in black, but the curve in orange.

A3-2: The left ordinate "Permeance" has been changed to the orange color

R3-3: Figure 3a: "H₂ Permeance", but Fig. 4 "H₂ permeability"

A3-3: "H₂ Permeance" has now changed to "H₂ permeance" in **Fig. 3a** and **3c**.

REVIEWERS' COMMENTS

Reviewer #1 (Remarks to the Author):

I am satisfied with the revised manuscript. Authors replied to my comments accordingly. Since the authors have well addressed my main concern about work originality by comparing with previous publications, this manuscript now is recommended to be accepted in Nature communication. And also, the separation mechanism of H₂/CO₂ through the CMS hollow fiber membrane synthesized in this study have been well discussed. The manuscript now is well writing with good characterizations and performances."

Hui-Hsin Tseng

RE: Point-by-point response for manuscript NCOMMS-20-18017B

Title: Carbon hollow fiber membranes for a molecular sieve with precise-cutoff ultramicropores for superior hydrogen separation

Authors: Linfeng Lei, Fengjiao Pan, Arne Lindbråthen, Xiangping Zhang, Magne Hillestad, Yi Nie, Lu Bai, Xuezhong He, Michael D. Guiver

Response to Reviewers and Editor comments:

The authors would like to thank the reviewers and the editor for their constructive comments. We have carefully considered all the editorial requests. To aid in the reviewing process, we have replied to all the requests on a point-by-point basis in the checklist file and highlighted the revised sections of the main manuscript with track changes. We hope the revised version can now be accepted for publishing in *Nature Communications*.

On behalf of the authors, and with kind regards,

Dr. Michael D. Guiver (designated corresponding author for the submission)

Dr. Xuezhong He (co-corresponding author)

Reviewer #1 (Remarks to the Author):

Reviewer: I am satisfied with the revised manuscript. Authors replied to my comments accordingly.

Since the authors have well addressed my main concern about work originality by comparing with previous publications, this manuscript now is recommended to be accepted in Nature communication. And also, the separation mechanism of H₂/CO₂ through the CMS hollow fiber membrane synthesized in this study have been well discussed. The manuscript now is well writing with good characterizations and performances.

Author Answer: The authors highly appreciate the encouraging comments. We also thank the reviewer for the recommendation of publishing our work in Nature Communications.